# Identifying key amino acid types that distinguish paralogous proteins using Shapley value based feature subset selection

## Abstract

Paralogous proteins have a common ancestor but have diverged in functionality. Using known machine learning algorithms, we present a data-driven method to identify the key amino acid types that play a role in distinguishing a given pair of proteins that are paralogs. We use an existing Shapley value based feature subset selection algorithm, SVEA, to identify the key amino acid types adequate to distinguish pairs of paralogous proteins. We refer to these as the amino acid feature subset ($AFS$). For a paralog pair, say proteins $P$ and $Q$, its $AFS$ is partitioned based on protein-wise importance as $AFS(P)$ and $AFS(Q)$ using a linear classifier, SVM. To validate the significance of the $AFS$ amino acids, we use multiple domain knowledge based methods : (a) multiple sequence alignment, and/or (b) 3D structure analysis, and/or (c) supporting evidence from biology literature. This method is computationally cheap, requires less data and can be used as an initial data-driven step for further hypothesis-driven experimental study of proteins. We demonstrate the results for 15 pairs of paralogous proteins. Code available at `https://anonymous.4open.science/r/AFS_AAC_SVM-F3D9`.

## 1 Introduction

Proteins form the fundamental machinery in living systems, having several vital functions such as DNA replication, catalysis, transport, environmental interaction, etc. Advancements in sequencing technologies have resulted in exponential growth of protein sequence databases (The UniProt Consortium, 2020). However, the number of experimentally verified annotations constitute a tiny fraction: only 0.57% of 250 million sequences in UniProtKB (The UniProt Consortium, 2020) have manually reviewed annotations. Experimental methods for determining biological process level functions (transcription, DNA repair, etc.) are high-throughput whereas methods for molecular function (catalysis, ligand specificity, etc.) are low-throughput and hence are not scalable. The relationship between sequence and function is subtle and has not been fully decoded yet.

Paralogs are proteins that have a common ancestor but have diverged functionally. The functional difference in two paralogous proteins is considered to arise due to evolutionary changes in the sequences (Yang et al., 2023). A typical experiment to investigate the role of an (or a group of) amino acid(s) in the function of a protein is to perform a site-directed mutagenesis experiment: replace one or more amino acids and test the effect of the sequence change (Kresge et al., 2006). In this work, we provide an algorithmic ML pipeline, consisting both feature engineering and feature subset selection, as a quick and resource-cheap test to assess the likely outcome from a site-directed mutagenesis experiment. Using this ML pipeline we investigate the following two questions:

- Does the coarse-grained information of amino acid composition carry a strong enough signal for discriminating paralogs?
- If so, which of the 20 amino acids have a potential role in the functional difference of the paralogs?

We use a diverse dataset of 15 paralog pairs. Our datasets show a range of sequence and function diversity (details in Appendix B). Longest common subsequence score ($lcss$) is a metric to quantify sequence diversity and median within-class $lcss$ is $\leqslant 0.5$ in 12 of the 15 datasets, and the median inter-class $lcss$ for the corresponding classes is less than within-class $lcss$. Functional diversity, as discerned from biology literature, also shows large diversity from subtle functional differences (e.g., trypsin/chymotrypsin) to drastic (e.g.,

lysozyme c/$\alpha$-lactalbumin). Function description is fine-grained (e.g., trypsin/chymotrypsin) as well as coarse grained (e.g, GPCRs).

Our findings are that small subsets of amino acids can discern differences between pairs of paralogs. The subset sizes are between 5 to 10, the median being 8. However, an established ground truth feature subset/rankings is not available for the task that can be used for evaluating the method or comparing performances with alternative methods. For this reason, we rely on evidence from experimental biology literature that highlights the role of the AFS amino acids in the function/structure of the respective protein. We provide validations from literature, MSA (a popular computational tool to assess evolutionary conservation) and logical consistencies; for many pairs such validations are more than one.

Towards this, we view a protein as the composite of its constituent standard 20 amino acids. We use amino acid composition (AAC) features, a Shapley value (Shapley, 1953) based feature subset selection algorithm (Shapley Value based Error Apportioning, SVEA) (Tripathi et al., 2020; 2021), and a linear support vector machine (SVM) classifier (Steinwart & Christmann, 2008) as tools to identify key amino acid types that can distinguish a given a pair of proteins that are paralogs. It yields quick results based on which biologists can conduct detailed experiments which are resource-intensive (time, cost, trained manpower, etc.).

The key results from our ML pipeline experiments are:

• Using known machine learning algorithms we demonstrate a data-driven method to identify key amino acids that distinguish two paralogous proteins.

- The SVEA algorithm identifies a subset of amino acid types (referred to as $AFS$) adequate for distinguishing two paralogous proteins. The size of $AFS$ ranges from 5 to 10 amino acids out of 20. (Table 1)
- For a paralog pair, say protein families $P$ and $Q$, the computed $AFS$ is partitioned into $AFS(P)$ and $AFS(Q)$ using a linear SVM, to determine the family-wise importance of $AFS$. (Table 1)

• Domain knowledge based validation of $AFS$: The significance of the amino acids in $AFS$ was validated for 14 datasets using various methods like (a) multiple sequence alignment (MSA) and/or (b) structural analysis and/or (c) supporting evidence from literature that report structural/functional role of these amino acids.

• Logical consistencies in the pair-wise $AFS$ of three paralogous proteins (globins, Section 3.1.7, and GPCRs, Section 3.1.8). If families $P$ vs $Q$ and $P$ vs $R$ have $AFS_1$ and $AFS_2$, then,

- we find common amino acids in $AFS_1(P)$ and $AFS_2(P)$, except for one pair.
- amino acids in $AFS_1 \cap AFS_2$ are either excluded from $AFS_3$, which is from $Q$ vs $R$, or have much lower Shapley value in $AFS_1$, $AFS_2$, or $AFS_3$.

• Validation of $AFS$ using test data (Section 3.2): The composition of amino acids is sufficient to classify several paralog pairs. A linear SVM classifies with high test scores (70-99%) using only the composition of $AFS$ amino acids as features. (Appendix Table 5)

• $AFS$ are top ranked features with an alternate feature ranking measure, Marginal Contribution feature importance (MCI) (Catav et al., 2021). (Appendix Table 6)

Shapley values based feature attribution methods are popular for explaining machine learning models (Rozemberczki et al., 2022). One such method is SHAP (Lundberg & Lee, 2017), which assigns attribution scores to input features based on a model's output for a given instance input. Another method is SAGE (Covert et al., 2020), which assigns feature attribution scores based on a model's loss computed at the dataset level. Unlike these methods, where feature attributions are based on a trained model, the SVEA algorithm that we use for our task assigns scores to the features based on the distribution of the data points in the feature space and their ground truth labels. The SVEA algorithm uses a function $v(S)$, which acts as a measure of inter-class linear separation between the data points in the space of the feature subset $S$. The scores assigned to the features are Shapley values computed using this function $v(\cdot)$. We also use an alternate feature ranking method, i.e. the Marginal Contribution Feature Importance (MCI) (Catav et al., 2021). MCI is an axiomatic approach that was proposed as an alternative to Shapley values to score and rank features. We find close agreement between the $AFS$ computed using SVEA and the top-ranked amino acids using MCI.

Use of deep learning methods trained on large datasets is becoming commonplace in Biology; for example, prediction of molecular function via EC number or GO annotation (Bileschi et al., 2022; Sanderson et al., 2023),

identifying input sequence regions relevant to model output (Zhou et al., 2016) and learning sequence-function mapping from deep mutational scanning experiment data (Song et al., 2021). The use of large datasets for training makes this approach highly resource-intensive. The approach we present herein needs much smaller datasets and, consequently, (i) is computationally cheap and (ii) has far wider applicability since labelled data validated by wet lab experiments is limited.

## 2  Methodology

We discuss the main components of our methodology.

### 2.1  AAC features

Consider a paralogous pair of proteins, families $P$ and $Q$. We first curate a set of sequences, say $D_P$ and $D_Q$, from a standard protein sequence database, SwissProt (The UniProt Consortium, 2020), with $n_P$ and $n_Q$ number of sequences each from families $P$ and $Q$ respectively. For a protein sequence $\mathbf{p}^{(j)} = (p_1^{(j)}, p_2^{(j)}, \ldots, p_L^{(j)})$ of length $L$ with $p_k^{(j)} \in \{1, 2, \cdots, 20\}$ corresponding to the standard 20 amino acids, the AAC feature $\mathbf{x}_j^{AAC} \in [0,1]^{20}$ for $\mathbf{p}^{(j)}$ is computed as follows,

$$x_{j,i}^{AAC} = \frac{1}{L} \sum_{k=1}^{L} \mathbf{1}_{\{p_k^{(j)}=i\}}, \ \forall i \in [20]$$

So $x_{j,i}^{AAC}$ is the normalised count of the standard amino acid $i$, $i \in \{1, 2, \cdots, 20\}$, in a protein $\mathbf{p}^{(j)}$.

### 2.2  Feature subset selection using SVEA

Given a set, $N$, of features from the protein sequences of $P$ and $Q$, we try to find the features $S \subseteq N$ that contribute the most to the linear separation of $P$ and $Q$ sequences. With $AAC$ features, we have $N = \{1, 2, \ldots, 20\}$ corresponding to each of the standard 20 amino acid types.

We utilise the Shapley value based feature ranking and subset selection algorithm, SVEA (Tripathi et al., 2020; 2021), to identify the most important feature subset $S \subseteq N$. Shapley value is a well known solution concept from cooperative game theory (Shapley, 1953; Narahari, 2014) for distributing the total worth of a coalition of players fairly among each of them by quantifying each players effective marginal contribution. The SVEA algorithm considers the binary classification task as a cooperative game among the features, with a function $v(S)$ as the worth of every feature subset $S$. $v(S)$ acts as a measure of linear separation between the classes in the feature space of $S$. Accounting for class-imbalance, we define $v(S)$ using a class-balanced hinge loss function $tr\_er(S)$, which is defined as,

$$tr\_er(S) = \min_{w, \xi_j} \frac{1}{2n_P} \sum_{j=1}^{n_P} \xi_j + \frac{1}{2n_Q} \sum_{j=n_P+1}^{n_Q} \xi_j$$

$$\text{s.t. } y_j \left( \sum_{i \in S} w_i x_{j,i}^{AAC} + b \right) \geq 1 - \xi_j, \ \forall j \in [n_P + n_Q]$$

$$\xi_j \geq 0, \ \forall j \in [n_P + n_Q]$$

and $v(S) = tr\_er(\varnothing) - tr\_er(S)$. The minimizer in the above finds a linear hyperplane with the least class-balanced hinge loss in the feature space of $S$. $\varnothing$ is the empty set and $tr\_er(\varnothing) = 1$, therefore, $v(S) = 1 - tr\_er(S)$. $tr\_er(S) = 0$ implies $v(S) = 1$, i.e., the two classes are completely linearly separable in the feature space of $S$. The maximum value of $tr\_er(S)$ possible is 1.

The Shapley value $\phi(i)$ for a feature $i \in N$ is computed as,

$$\phi(i) = \sum_{S \subseteq N \setminus \{i\}} \frac{|S|!(|N| - |S| - 1)!}{|N|!} (v(S \cup \{i\}) - v(S)).$$

Thus, $\phi(i)$ is a weighted sum of the marginal contribution of feature $i$ to all the possible feature subsets that do not contain $i$. Shapley values are unique solution concepts satisfying the axioms - efficiency, symmetry and marginality (Young, 1985). The higher the $\phi(i)$, the higher the contribution of feature $i$ to the linear separation between the classes and, consequentially, the higher the importance of feature $i$ distinguishing the classes.

Exact Shapley value computations are known to be exponential time. Hence, they are computed using a linear time (in number of features) Monte Carlo approximation (Castro et al., 2009) in the SVEA algorithm. As the number of features is small (20), good approximations can be computed fast via larger sampling. Furthermore, we do not find significant variance in the Monte Carlo estimates across multiple runs. We also find that the amino acids with top Shapley values are consistent across these runs (see Appendix E.4 for a detailed variance analysis). More details of the SVEA algorithm are given in Appendix Section C.

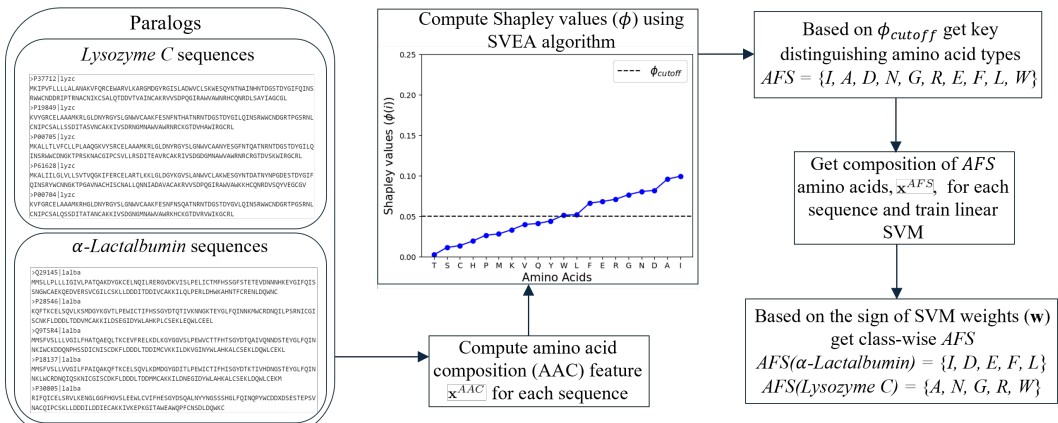

Figure 1: Flowchart summarizing the steps in our ML pipeline to compute the key amino acid types, $AFS$, that distinguish two paralogous proteins, using amino acid composition (AAC) features, Shapley value based SVEA algorithm for feature subset selection and class-wise feature subsets using linear SVM. Lysozyme C and $\alpha$-Lactalbumin are used here as representative examples of paralog pairs. $AFS$ identified for other paralog pairs are given in Table 1.

**Data-driven cutoff for selecting** $AFS$: The efficiency axiom of Shapley value implies, $\sum_{i=1}^{20} \phi(i) = v(N)$. If all features have equal contribution in achieving $v(N)$, then $\phi(i) = \frac{v(N)}{20}, \forall i \in N$. Consequentially, if a feature $i$ had lesser contribution than others then $\phi(i) < \frac{v(N)}{20}$. Therefore, we set $\phi_{cutoff} = \frac{v(N)}{20}$ for selecting the key distinguishing amino acid feature subset, $AFS = \{i : \phi(i) \geqslant \phi_{cutoff}\}$. Each of the features in $AFS$ uniquely corresponds to $d \leqslant 20$ amino acids from the standard 20.

### 2.3 Protein family-wise partition of $AFS$ using SVM

We train a linear SVM, to classify $P$ vs $Q$, using the composition of the amino acids in $AFS$ as the features, i.e. using $\mathbf{x}_j^{AFS} \in [0,1]^d$, with $x_{j,i'}^{AFS} = x_{j,i}^{AAC}$ and each $i' \in \{1, 2, \cdots, d\}$ uniquely maps to a $i \in AFS$. We use these linear SVM weights $\mathbf{w} \in \mathbb{R}^d$ to divide the set $AFS$ into disjoint sets $AFS(P)$ and $AFS(Q)$ based on the sign of the weights. Since $x_{j,i'}^{AFS} \geqslant 0 \ \forall i' \in [d]$, the sign of the linear classifier weight $w_{i'}$ indicates which class is relatively prominent in the amino acid corresponding to $i'$. So if the $+1$ class is $P$, then we divide $AFS$ class-wise as $AFS(P) = \{i' \in [d] : w_{i'} > 0\}$ and similarly $AFS(Q) = \{i' \in [d] : w_{i'} < 0\}$. See Appendix Section D for details on SVM training.

A flowchart summarizing the steps for computing $AFS(P)$ and $AFS(Q)$ is shown in Figure 1.

### 2.4 Validation of $AFS$

**Literature evidence**: For 14 different paralog protein pairs, we provide supporting evidence from protein biology literature for the significance of amino acids in $AFS$ in the functional specificity of the protein pair.

**MSA analysis**: We also compute multiple sequence alignment (MSA) of randomly selected sequences from $D_P$ and $D_Q$ and analyze the conservation of $AFS(P)$ and $AFS(Q)$ amino acids within and across the respective families (Figure 2). MSA algorithms (Edgar & Batzoglou, 2006) aim to align multiple protein sequences by inserting gaps in the sequences while optimizing an objective. The objective is usually to minimize the number of gaps inserted while maximizing an overall score that promotes the alignment of similar (based on physicochemical properties) amino acids at a given position. The alignments are often used as a tool to determine homologous relationships between proteins and identify conserved or mutated regions in them.

**Structural analysis**: For paralog pairs that together function as heteromers (protein complexes made up of different types of proteins), we perform structural analysis to validate the role of $AFS$ in the heteromeric structure formed by the paralog pair (Sections 3.1.7, 3.1.3 and 3.1.4).

**Using test data**: We test the classifier trained in Section 2.3 on a test data. (Details on test data in Appendix Section A.1). In general, we find an imbalance in the number of sequences for the two paralogous proteins. It is known that accuracy is not a well-suited performance measure of the classifier in class imbalance settings. Therefore, we use the arithmetic mean of sensitivity and specificity (AM) to measure the performance of the classifier (Brodersen et al., 2010).

**Using marginal contribution feature importance (MCI)**: We check agreement of $AFS$ with another feature ranking method, MCI (Catav et al., 2021). See Appendix Section E.3 for details on MCI computation.

## 3 Results and Discussions

### 3.1 Role of the amino acids identified in $AFS$

For 15 paralog pairs, we discuss the significance of the amino acids identified in the respective $AFS$ (Table 1).

#### 3.1.1 Lysozyme C and $\alpha$-Lactalbumin

**Literature evidence**: Amino acids $D$ and $E$ of $AFS(\alpha$-Lactalbumin$)$ are found in the $Ca^{2+}$ and $Zn^{2+}$ binding sites respectively of $\alpha$-lactalbumin (Permyakov & Berliner, 2000; Permyakov, 2020). All $\alpha$-lactalbumins studied so far are known to bind $Ca^{2+}$ and $Zn^{2+}$ whereas several (but not all) lysozymes do not bind $Ca^{2+}$.

**MSA analysis**: (Figure 2a) $AFS(\alpha$-Lactalbumin$)$ and $AFS($Lysozyme C$)$ amino acids (Table 1) are significantly conserved in respective families.

#### 3.1.2 Trypsin and Chymotrypsin

**Literature evidence**: $Y$ and $W$ get the highest Shapley value $\phi(\cdot)$ in $AFS($Trypsin$)$ and $AFS($Chymotrypsin$)$ respectively (Table 1 and Figure 7b). In experiments to convert trypsin to chymotrypsin (Hedstrom et al., 1994; Hedstrom, 2002) it has been shown that $Y$ to $W$ conversion in loop-3 of trypsin leads to significant increase in chymotrypsin activity. We do not find $S, H$ and $D$ in $AFS$, which are important for the function of both families and are known as the catalytic triad (Dodson & Wlodawer, 1998).

#### 3.1.3 Tubulin-$\alpha$ and Tubulin-$\beta$

**MSA analysis**: (Appendix Figure 10) $AFS($Tubulin-$\alpha)$ and $AFS($Tubulin-$\beta)$ amino acids are significantly conserved in respective families.

**Structural analysis of $AFS$**: Tubulins typically exist as heterodimers, consisting of two subunits: tubulin-$\alpha$ and tubulin-$\beta$ (Mühlethaler et al., 2021). We looked at the contact residues of a tubulin-$\alpha$ chain and tubulin-$\beta$ chain in the 3D structure of tubulin-$\alpha/\beta$ heterodimer (PDB IDs: 3JAR, 5N5N). We see that the contact points

Table 1: $AFS$ and its class-wise partition computed for 15 paralog pairs. The number of unique sequences from the SwissProt (The UniProt Consortium, 2020) database used for computing $AFS$ is given inside parenthesis ($\cdot$) for each protein family. Data collection details are in Appendix Section A.1. $AFS$ amino acids are written in decreasing Shapley values from left to right for each paralog pair. Figures 3 and 7 show the Shapley value of the amino acids for each paralog pair. For globins and GPCRs, common acids across different $AFS$ within a paralog triplet are colour-coded.

| Paralog pair | Amino acid feature subset, $AFS$ | Class-wise $AFS$ parition |
|:---:|:---:|:---:|
| Lysozyme C (74) and $\alpha$-Lactalbumin (22) | $\{I, A, D, N, G, R, E, F, L, W\}$ | $AFS(\alpha\text{-Lactalbumin}) = \{I, D, E, F, L\}$ 
 $AFS(\text{Lysozyme C}) = \{A, N, G, R, W\}$ |
| Trypsin (66) and Chymotrypsin (17) | $\{Y, W, T, A, V, K, P\}$ | $AFS(\text{Trypsin}) = \{Y, A\}$ 
 $AFS(\text{Chymotrypsin}) = \{W, T, V, K, P\}$ |
| Tubulin-$\alpha$ (117) and Tubulin-$\beta$ (191) | $\{M, Q, K, N, F, I, H, A, C, Y\}$ | $AFS(\text{Tubulin-}\alpha) = \{K, I, H, C, Y\}$ 
 $AFS(\text{Tubulin-}\beta) = \{M, Q, N, F, A\}$ |
| Histone H2A (180) and Histone H2B (177) | $\{L, G, S, M, K, N, T, Y, F\}$ | $AFS(\text{Histone H2A}) = \{L, G, N\}$ 
 $AFS(\text{Histone H2B}) = \{S, M, K, T, Y, F\}$ |
| Interleukin-1 $\alpha$ (16) and Interleukin-1 $\beta$ (25) | $\{C, G, T, S, V, Q, A, N, P\}$ | $AFS(\text{Interleukin-1 }\alpha) = \{T, S, A, N\}$ 
 $AFS(\text{Interleukin-1 }\beta) = \{C, G, V, Q, P\}$ |
| Cytochrome P450 CYP3 (32) and CYP51 (32) | $\{H, F, G, K, A, P, N\}$ | $AFS(\text{CYP3}) = \{F, K, P, N\}$ 
 $AFS(\text{CYP51}) = \{H, G, A\}$ |
| **Globins** | | |
| Myoglobin (107) and Hemoglobin-$\alpha$ (303) | $AFS_1 = \{E, S, Y, V, K, P, I, G, C, W\}$ | $AFS_1(\text{Myoglobin}) = \{E, K, I, G, W\}$ 
 $AFS_1(\text{Hemoglobin-}\alpha) = \{S, Y, V, P, C\}$ |
| Myoglobin (107) and Hemoglobin-$\beta$ (285) | $AFS_2 = \{K, V, C, E, W, N, F, M, Y, I\}$ | $AFS_2(\text{Myoglobin}) = \{K, E, M, I\}$ 
 $AFS_2(\text{Hemoglobin-}\beta) = \{V, C, W, N, F, Y\}$ |
| Hemoglobin-$\alpha$ (303) and Hemoglobin-$\beta$ (285) | $AFS_3 = \{W, P, N, S, G\}$ | $AFS_3(\text{Hemoglobin-}\alpha) = \{P, S\}$ 
 $AFS_3(\text{Hemoglobin-}\beta) = \{W, N, G\}$ |
| **GPCRs** | | |
| Rhodopsin-like (181) and Glutamate-like (89) | $AFS_1 = \{D, Q, E, G, M, L\}$ | $AFS_1(\text{Rhodopsin}) = \{M, L\}$ 
 $AFS_1(\text{Glutamate}) = \{D, Q, E, G\}$ |
| Secretin-like (90) and Glutamate-like (89) | $AFS_2 = \{W, H, Y, V, D\}$ | $AFS_2(\text{Secretin}) = \{W, H, Y\}$ 
 $AFS_2(\text{Glutamate}) = \{V, D\}$ |
| Rhodopsin-like (181) and Secretin-like (90) | $AFS_3 = \{W, E, M, S, V, H, Q, A\}$ | $AFS_3(\text{Rhodopsin}) = \{M, S, V, A\}$ 
 $AFS_3(\text{Secretin}) = \{W, E, H, Q\}$ |
| **Rhodopsin-like GPCRs** | | |
| Aminergic receptors (186) and Lipid receptors (113) | $AFS_1 = \{L, P, E, W, F, M, D\}$ | $AFS_1(\text{Aminergic receptors}) = \{P, E, W, D\}$ 
 $AFS_1(\text{Lipid receptors}) = \{L, F, M\}$ |
| Aminergic receptors (186) and Peptide receptors (367) | $AFS_2 = \{L, F, E, M, K, D, V, R\}$ | $AFS_2(\text{Aminergic receptors}) = \{E, K, D, R\}$ 
 $AFS_2(\text{Peptide receptors}) = \{L, F, M, V\}$ |
| Lipid receptors (113) and Peptide receptors (367) | $AFS_3 = \{P, R, G, I, W, S, V\}$ | $AFS_3(\text{Lipid receptors}) = \{R, G, S\}$ 
 $AFS_3(\text{Peptide receptors}) = \{P, I, W, V\}$ |

of the tubulin-$\alpha$ chain in the heterodimer have more $AFS(\text{Tubulin-}\alpha)$ amino acids than $AFS(\text{Tubulin-}\beta)$. Similarly, $AFS(\text{Tubulin-}\beta)$ amino acids are more than $AFS(\text{Tubulin-}\alpha)$ at the contact point of the tubulin-$\beta$ chain in the heterodimer. Thus, the amino acids identified in $AFS$ can be considered to be significant towards the quaternary structure of tubulin-$\alpha/\beta$ heterodimer. Appendix Section E.1 has more details.

### 3.1.4 Histone H2A and Histone H2B

**MSA analysis**: (Appendix Figure 11), $AFS(\text{Histone H2A})$ and $AFS(\text{Histone H2B})$ amino acids are significantly conserved in respective families.

**Structural analysis of** $AFS$: Histones have a heterooctameric structure comprising of two H2A/H2B dimers and one H3/H4 tetramer (Dutta et al., 2001). We looked at the contact residues of an H2A chain and H2B chain in the heteroocatmer structure of histone (PDB IDs: 3KWQ, 1AOI). We find that the contact points of H2A chain in the heterooctamer have more $AFS$(Histone H2A) amino acids than $AFS$(Histone H2B). This is interesting since $AFS$(Histone H2A) has only three amino acids, while $AFS$(Histone H2B) has six amino acids. Similarly, the contact points of H2B chain in the heterooctamer have more $AFS$(Histone H2B) amino acids than $AFS$(Histone H2A). Thus, the amino acids identified in $AFS$ can be considered to be significant towards the quaternary structure of the histone heterooctamer. See Appendix Section E.2 for more details.

### 3.1.5   Interleukin-1 $\alpha$ and Interleukin-1 $\beta$

**Literature Evidence:** $C$ has the highest Shapley value and is in $AFS$(Interleukin-1 $\beta$). Deleting $C$ results in loss of activity in Interleukin-1 $\beta$ (Veerapandian et al., 1992). We do not find such studies for Interleukin-1 $\alpha$.

**MSA analysis**: (Appendix Figure 12) $AFS$(Interleukin-1 $\alpha$) and $AFS$(Interleukin-1 $\beta$) amino acids show significant conservation in respective families.

### 3.1.6   Cytochrome P450 CYP3 and CYP51

**Literature evidence**: $H, F$ and $G$, in the respective order, have the highest Shapley value $\phi(\cdot)$ for this paralogous pair (Table 1 and Figure 7f). $H$ and $G$ with the highest $\phi(\cdot)$ in $AFS$(CYP51) have been reported (Nitahara et al., 2001; Lepesheva & Waterman, 2004; 2007; Strushkevich et al., 2010) to be important in the enzymatic activity of CYP51. Mutation of these amino acids at specific positions has been shown to result in a decrease in the activity of the enzyme (Lepesheva & Waterman, 2007; 2004). Similarly, $F$ with the highest $\phi(\cdot)$ in $AFS$(CYP3) is also known to be important in the enzymatic activity of CYP3 (Qiu et al., 2008; Denisov et al., 2019; Zhang et al., 2024). A cluster of $F$ residues in CYP3 is known to form a substrate-binding pocket with an active site (Zhang et al., 2024).

### 3.1.7   Globins

**MSA analysis:** (Figures 2b,2c and Appendix Figure 9) For the three globin paralog pairs (Table 1), we observe in the MSA, conservation of the class-wise partition of $AFS$ in the respective families.

**Structural analysis of** $AFS$: Myoglobin is a monomer, while $\alpha$ and $\beta$ chains together constitute hemoglobin, a tetramer of composition $\alpha_2\beta_2$ (Dill et al., 2017). We superimposed the 3D structures of myoglobin, hemoglobin-$\alpha$ and hemoglobin-$\beta$ (PDB IDs: 3RGK, 1HHO) and mapped the $\alpha, \beta$ contact residues (based on (Shionyu et al., 2001)) of hemoglobin tetramer to that of myoglobin. We find that the amino acids $K, E, I$, which are common in $AFS_1$(Myoglobin) and $AFS_2$(Myoglobin), are less in number at the contact residues of hemoglobin tetramer and more in number at the corresponding locations in myoglobin, which is a monomer (see Figure 4).

**Literature evidence:**  $W$ with a significantly high Shapley value $\phi(W)$ (Figure 3b), is present in $AFS_3$(Hemoglobin-$\beta$). It is highly conserved at position 40 in the MSA (Figure 2c) in hemoglobin-$\beta$ sequences as compared to hemoglobin-$\alpha$ sequences. This $W$ at position 40 has been determined to be present in hemoglobin-$\beta$ at one of its contact positions to hemoglobin-$\alpha$ in the tetrameric structure (Shionyu et al., 2001) and is, therefore, a structurally and functionally significant residue. $C$, present in $AFS_1$(Hemoglobin-$\alpha$) and $AFS_2$(Hemoglobin-$\beta$), has been shown to play an important role in the tetrameric structure of hemoglobin formed by $\alpha$ and $\beta$ hemoglobins (Kan et al., 2013).

**Logical consistencies in** $AFS$ (refer to Table 1 (Globins) for $AFS_1, AFS_2, AFS_3$):

• $AFS_1 \cap AFS_2 = \{E, Y, V, K, I, C, W\}$. Except for $W$ with the least Shapley value in $AFS_1$ (Figure 3a), the remaining are excluded from $AFS_3$.

  • *Explanation*:$V, Y, C$ in $AFS_1$(Hemoglobin-$\alpha$) $\cap$ $AFS_2$(Hemoglobin-$\beta$) can be expected not to be *key* in $\overline{AFS_3}$ for distinguishing $\alpha$ vs $\beta$ hemoglobin.

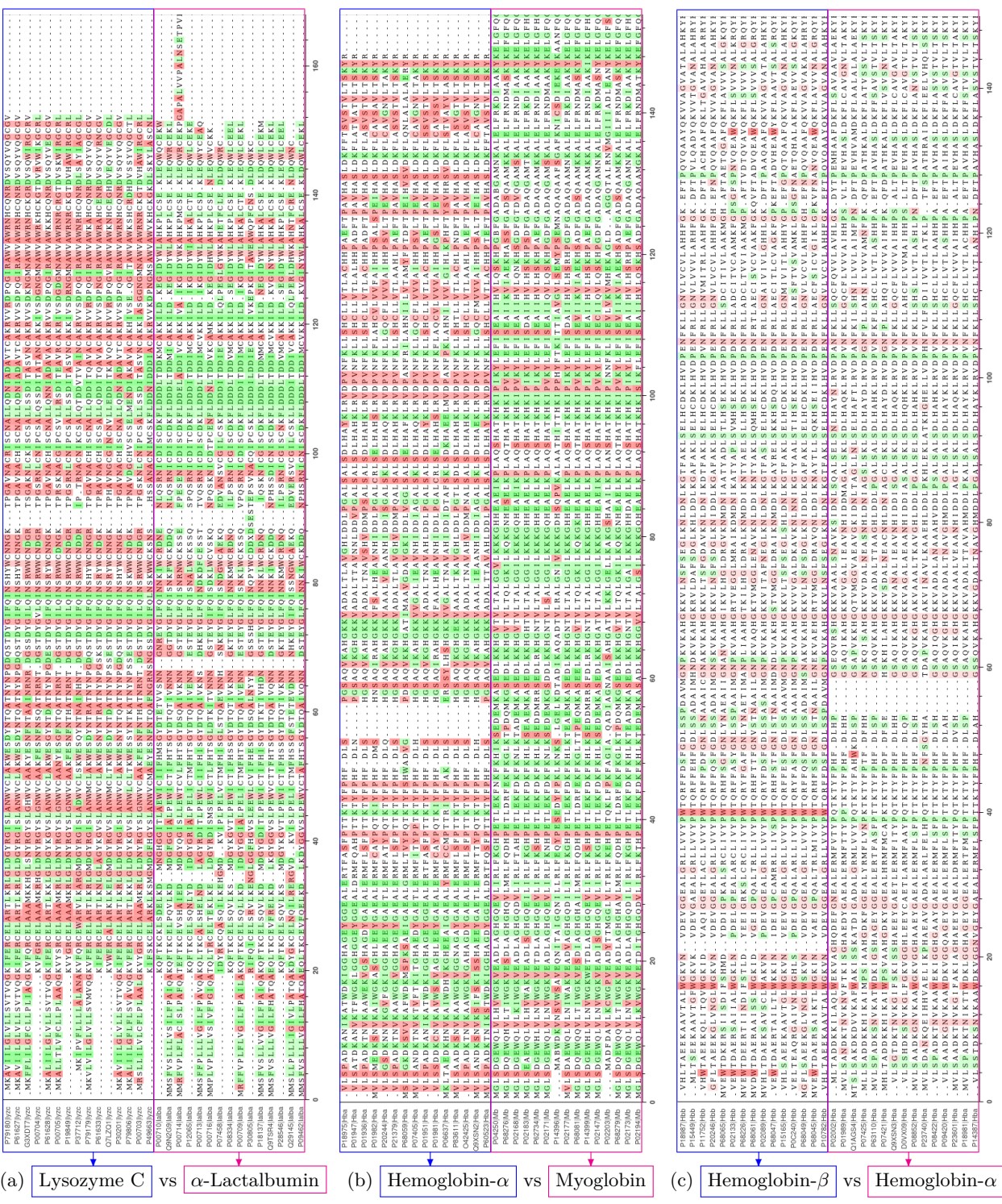

Figure 2: Multiple sequence alignment of sequences from the respective families in (a), (b) and (c). Within each alignment, 15 sequences on the left are from one family, and those on the right are from the other family in each of (a), (b) and (c). The sequences are randomly selected from the train set of the families. For each aligned sequence in (a) $AFS(\alpha\text{-Lactalbumin})$ amino acids are in green and $AFS(\text{Lysozyme C})$ are in red, in (b) the amino acids in $AFS_1(\text{Myoglobin})$ are in green and $AFS_1(\text{Hemoglobin-}\alpha)$ are in red, and in (c) the amino acids in $AFS_2(\text{Hemoglobin-}\alpha)$ are in green and $AFS_2(\text{Hemoglobin-}\beta)$ are in red. The intensity of the color is proportional to the Shapley value $\phi(i)$ of the amino acid $i$ (Figures 3 and 7).

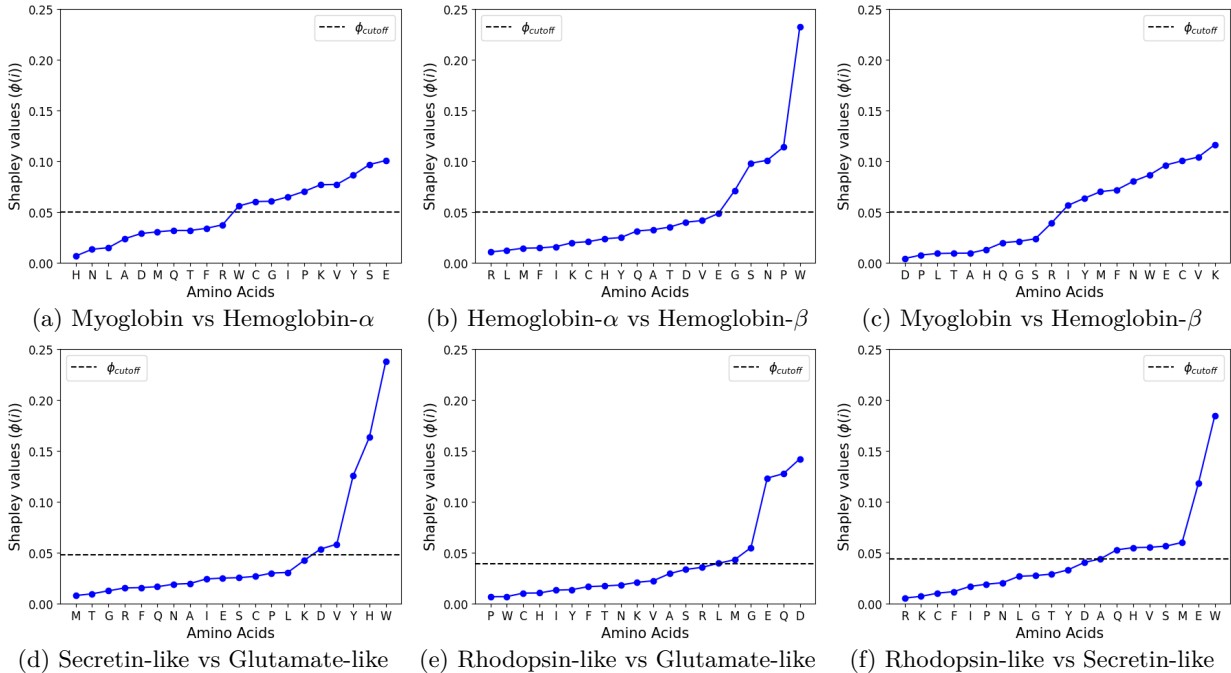

Figure 3: Shapley value ($\phi(i)$) for AAC features computed using SVEA. See Appendix Figure 7 for remaining paralogs.

- $AFS_2 \cap AFS_3 = \{W, N\}$. $N$ is excluded from $AFS_1$, while $W$ gets the least Shapley value in $AFS_1$ (Figure 3a).

- $AFS_3 \cap AFS_1 = \{W, P, S, G\}$. $\{P, S, G\}$ are excluded from $AFS_2$, while $W$ gets the least Shapley value in $AFS_1$.

The Shapley value for $W$ is very close to the cut-off in $AFS_1$ (Figure 3a). If it is dropped from $AFS_1$, then the exclusion principle illustrated above would be more prominent as in GPCRs (Section 3.1.8).

### 3.1.8 G-protein coupled receptors (GPCRs)

**Literature evidence**: $W$ (with highest Shapley value $\phi(\cdot)$) and $H$ common in $AFS_2$(Secretin) and $AFS_3$(Secretin) (Table 1 and Figure 3), are well conserved at multiple positions with structural importance and functional importance in secretin-like GPCR sequences (Cary et al., 2022; Harmar, 2001). Mutating certain conserved $W$ leads to a loss in expression of this GPCR at the cell surface, where it functions (Cary et al., 2022). $H$ present in the intracellular loop region is also known to be important in the activation of certain secretin-like GPCRs (Harmar, 2001).

$M$ common in $AFS_1$(Rhodopsin) and $AFS_3$(Rhodopsin) has been found to be present at important binding pockets and a position important for activation of the GPCR (Okada et al., 2001; Sakmar et al., 2002). $S$ from $AFS_3$(Rhodopsin) is found at multiple major phosphorylation sites (see Okada et al. 2001 for details) in Rhodopsin.

Mutating $D$ at two positions has been shown to affect glutamate binding of glutamate receptor GPCRs (Jingami et al., 2003). $D$ is common in $AFS_1$(Glutamate) and $AFS_2$(Glutamate) and has highest Shapley value in $AFS_1$.

$E$ and $D$ common in $AFS_1$(Aminergic) and $AFS_2$(Aminergic) are present at binding sites of important ligands (like histamine/serotonin) of aminergic receptors (Vass et al., 2019).

**Logical consistencies in $AFS$ of GPCRs** (refer to Table 1 (GPCRs) for $AFS_1, AFS_2, AFS_3$):

```
3RGK.A/Myoglobin/1-40           GLSDGEWQLVLNVWGKVEADIPGHGQEVLIRLFKGHPETL
1HHO.A/Hemoglobin-α/1-40        VLSPADKTNVKAAWGKVGAHAGEYGAEALERMFLSFPTTK

3RGK.A/Myoglobin/41-80          EKFDRFKHLKSEDEMKASEDLKKHGATVLTALGGILKKKG
1HHO.A/Hemoglobin-α/41-74       TYFPHFDL------SHGSAQVKGHGKKVADALTNAVAHVD

3RGK.A/Myoglobin/81-120         HHEAEIKPLAQSHATKHKIPVKYLEFISEAIIQVLQSKHP
1HHO.A/Hemoglobin-α/75-114      DMPNALSALSDLHAHKLRVDPVNFKLLSHCLLVTLAAHLP

3RGK.A/Myoglobin/121-147        GDFGADAQGAMNKALELFRKDMASNYK
1HHO.A/Hemoglobin-α/115-141     AEFTPAVHASLDKFLASVSTVLTSKYR
```

(a) alignment of myoglobin (brown) and hemoglobin-$\alpha$ (blue) sequences based on structure alignment (right)

```
3RGK.A/Myoglobin/1-40           GLSDGEWQLVLNVWGKVEADIPGHGQEVLIRLFKGHPETL
1HHO.B/Hemoglobin-β/1-38        HLTPEEKSAVTALWGKV--NVDEVGGEALGRLLVVYPWTQ

3RGK.A/Myoglobin/41-80          EKFDRFKHLKSEDEMKASEDLKKHGATVLTALGGILKKKG
1HHO.B/Hemoglobin-β/39-78       RFFESFGDLSTPDAVMGNPKVKAHGKKVLGAFSDGLAHLD

3RGK.A/Myoglobin/81-120         HHEAEIKPLAQSHATKHKIPVKYLEFISEAIIQVLQSKHP
1HHO.B/Hemoglobin-β/79-118      NLKGTFATLSELHCDKLHVDPENFRLLGNVLVCVLAHHFG

3RGK.A/Myoglobin/121-147        GDFGADAQGAMNKALELFRKDMASNYK
1HHO.B/Hemoglobin-β/119-145     KEFTPPVQAAYQKVVAGVANALAHKYH
```

(b) alignment of myoglobin (brown) and hemoglobin-$\beta$ (silver) sequences based on structure alignment (right)

Figure 4: The hemologlobin-$\alpha/\beta$ tetramer contact points (as identified in Table 3 and Table 4 of Shionyu et al. (2001)) are highlighted in the hemoglobin sequences. These have also been highlighted in green in the structure alignments (right). The highlighted amino acids in the myoglobin chain correspond to the positions that align with hemologlobin-$\alpha/\beta$ tetramer contact points. The amino acids $K, E, I$, which are common in $AFS_1$(Myoglobin) and $AFS_2$(Myoglobin), are in **bold** in the alignment. We find that $K, E, I$ are less in number at the contact residues of hemoglobin tetramer and more in number at the corresponding locations in myoglobin, which is a monomer. Structure alignment is done using the online tool available at `https://www.rcsb.org/alignment`, with the default parameter settings (`|algorithm: jFATCAT(rigid)||RMSD Cutoff: 3||AFP Distance Cutoff: 1600||Fragment Length: 8|`).

- $AFS_1 \cap AFS_2 = \{D\}$, is excluded from $AFS_3$.

- $AFS_2 \cap AFS_3 = \{W, H, V\}$, is excluded from $AFS_1$.

- $AFS_3 \cap AFS_1 = \{Q, E, M\}$, is excluded from $AFS_2$.

**Logical consistencies in $AFS$ of Rhodopsin-like GPCR subfamilies** (refer to Table 1 (Rhodopsin-like GPCRs) for $AFS_1, AFS_2, AFS_3$):

- $AFS_1 \cap AFS_2 = \{L, E, F, M, D\}$, is excluded from $AFS_3$.

- $AFS_2 \cap AFS_3 = \{R, V\}$, is excluded from $AFS_1$.

- $AFS_3 \cap AFS_1 = \{P, W\}$ is excluded from $AFS_2$.

The explanations for these consistencies are similar to that in globins (Section 3.1.7).

## 3.2 Validation of $AFS$ using test data

The classification scores on test data for the classifiers trained using AAC and $AFS$ features, respectively, are reported in Appendix Table 5. Using $AFS$ features, the test AM scores are at least 70%. For 13 of 15 paralog pairs, the scores are greater than 83%, and for 8 of 15 paralog pairs, it is greater than 90%. Details of the test data are provided in Appendix Section A.1.

### 3.2.1 Test scores on random feature subsets

For each paralog pair, we train classifiers using 100 random feature subsets of the same size as $AFS$. We compute the mean test AM scores of these classifiers with the $AFS$-based classifier (See Appendix Table 7). The mean test AM score drops for these random feature subsets in comparison to the $AFS$-based score for all 15 paralog pairs. The mean test AM further drops for 14 pairs, if the randomly selected amino acids contain only non-$AFS$ amino acids. Details in Appendix Section E.5.

### 3.3 Marginal contribution feature importance (MCI) of $AFS$

For an $AFS$ of size $d$, the top-$d$ amino acids ranked by MCI differ with $AFS$ only in at the most two amino acids. For 8 of 15 datasets, $AFS$ and top-$d$ MCI sets are the same, while only for two datasets do they differ in two amino acids. For all 15 datasets, at least the top-3 MCI amino acids are in $AFS$. For 11 of these datasets, at least the top-5 MCI amino acids are in $AFS$. (Appendix Table 6)

### 3.4 Relating $AFS$ to diseases

We report some examples that link AFS amino acids to diseases. This is based on evidence from biology literature that suggests the role of these amino acids in the function/structure of the respective families. Thus, modification/mutation of these residues may be disease-causing as it could affect the structure and/or function of the protein, leading to disruption of one or more physiological processes.

**Example 1**: $W$ has the highest Shapley value in $AFS_2(\text{Secretin})$ and $AFS_3(\text{Secretin})$ (Table 1). Mutating certain conserved $W$ leads to a loss in cell-surface expression of calcitonin generelated peptide receptor (CGRPR), a secretin-like GPCR (Cary et al., 2022). A study (Ringer et al., 2017) in mice reports that disruption in CGRPR signalling accelerates Amyotrophic Lateral Sclerosis (ALS).

**Example 2**: ($AFS$ of hemoglobin-$\beta$ - from myoglobin vs hemoglobin-$\beta$ and hemoglobin-$\alpha$ vs hemoglobin-$\beta$) (Table 1)

(a) $W$ with a significantly high Shapley value $\phi(W)$ (Figure 3(b)), is present in $AFS_3(\text{Hemoglobin-}\beta)$. It is also in $AFS_2(\text{Hemoglobin-}\beta)$ (Table 1). $W$ to $S$ and $W$ to $R$ mutation at a certain position in hemoglobin-$\beta$ has been reported to result in abnormal hemoglobins - Hemoglobin Hirose (Sasaki et al., 1978) and Hemoglobin Rothschild (Sharma et al., 1980), respectively. These have altered oxygen affinities and dissociation of hemoglobin tetramer to dimers.

(b) $N$ is common in $AFS_3(\text{Hemoglobin-}\beta)$ and $AFS_2(\text{Hemoglobin-}\beta)$. Mutation/deletion of $N$ has been associated with $\beta$-thalassemia (Thein, 2013).

(c) $V$ has the highest Shapley value in $AFS_2(\text{Hemoglobin-}\beta)$ and mutations of this residue are related to hemoglobin variants with altered structure and biochemical properties leading to varying physiological effects (Thom et al., 2013). An example is Hemoglobin Olympia (Stamatoyannopoulos et al., 1973), having $V$ at a particular position mutated to $M$.

## 4 Conclusion

We demonstrated an ML pipeline to identify the key amino acid types, $AFS$, that distinguish a pair of paralogous proteins. The role of $AFS$ in functionally distinguishing the paralog pairs was validated using various sources of domain knowledge. The robustness of this approach, as demonstrated by considering a diverse set of paralogous protein pairs, illustrates its wider applicability. Identification of $AFS$ can be used as an initial data-driven step before doing more detailed experimental investigations, like site-directed mutagenesis (Bachman, 2013) resolving sequence-function relationship. Our experiments suggest that amino acids with high Shapley values are more likely to play a role in the functional difference between the paralogs and, hence, can be targeted for site-directed mutagenesis experiments. As the size of $AFS$ is small (5-10 amino acids of 20), significantly fewer mutations can be tried.

As our pipeline works without using the sequence order information of the amino acids in the protein, it posits an interesting question to biologists : how amino acid composition by itself is able to distinguish paralogs given ample evidence that 3D structure and function are conserved despite sequence divergence (Lau et al., 2015)! Notably, amino acids in the $AFS$ typically occur more than once in the sequence, but our method is silent on the specific positions where the amino acid has a functionally distinguishing role. This may be addressed by engineering features that incorporate sequence order information from the protein. However, these features can be very high-dimensional, for example, $20^k$-dimensional for $k$-mer features. The Monte Carlo based approximation algorithm for Shapley values would require exponentially more sampling (in number of features) for good approximations.

## Impact Statement

This paper presents a computationally efficient data lean ML pipeline. It can be used by biologists to decide whether they should invest valuable resources (skilled manpower, time, funds, etc.) for performing wet-lab experiments to determine amino acid(s) that are critical for functional differentiation of paralogous proteins.

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

# A    Data collection and code

We discuss the details of the data collection procedure for the datasets used in our computational experiments.

## A.1    Datasets of 15 paralog pairs

We apply our method for identifying amino acid types that distinguish paralogous proteins using the datasets described in Table 2. Only the train set is used for computing $AFS$, while the test set is used for computing classification scores for the linear SVM trained using the train set.

Table 2: The number of sequences in the train and test sets of the protein families considered in computational experiments.

| Family | Train (Swiss-Prot) | Test (TrEMBL) |
|---|---|---|
| **Lysozyme-like** | | |
| $\alpha$-Lactalbumin | 22 | 53 |
| Lysozyme C | 74 | 14 |
| **Trypsin-like** | | |
| Trypsin | 66 | 3813 |
| Chymotrypsin | 17 | 281 |
| **Tubulin** | | |
| $\alpha$ | 117 | 190 |
| $\beta$ | 191 | 347 |
| **Histone** | | |
| H2A | 180 | 16599 |
| H2B | 177 | 7599 |
| **Interleukin-1** | | |
| $\alpha$ | 16 | 12 |
| $\beta$ | 25 | 194 |
| **Cytochrome P450** | | |
| CYP3 | 32 | 818 |
| CYP51 | 32 | 601 |
| **Globins** | | |
| Myoglobin | 107 | 479 |
| Hemoglobin-$\alpha$ | 303 | 525 |
| Hemoglobin-$\beta$ | 285 | 261 |
| | (GPCR-PEnDB) | |
| | Train (80%) | Test (20%) |
| **GPCR families** | | |
| Rhodopsin-like | 181 | 45 |
| ↳ Lipid receptors | 113 | 28 |
| Peptide receptors | 367 | 92 |
| Aminergic receptors | 186 | 47 |
| Glutamate-like | 89 | 23 |
| Secretin-like | 90 | 23 |

All datasets are taken from publicly available databases (UniProt (The UniProt Consortium, 2020) and GPCR-PEnDB (Begum et al., 2020)). Well-known pairs of paralogous proteins were curated from millions of sequences from UniProt considering the number of sequences and manually reviewed labels available for them.

For all datasets except GPCR, we use manually curated Swiss-Prot sequences for training and electronically annotated TrEMBL sequences for testing. These proteins have very specific functions. In contrast, GPCRs are a large and diverse group of transmembrane proteins that mediate cellular responses to extracellular signals. We chose to use an already curated dataset in this case. For each of the GPCR families considered (Table 2), the sequences are randomly split as 80%-train/20%-test. The use of GPCR-PEnDB data is to illustrate the effectiveness of our method with random slicing, which is inevitable when additional curated data are not available. If one or many UniProt entries in a dataset had identical sequences, then only one of them was retained, and the remaining were deleted.

The following queries were used for collecting data from UniProt (The UniProt Consortium, 2020),

- **lysozyme C**: `(protein_name:"lysozyme C") AND (fragment:false) NOT (existence:4) NOT (existence:5) AND (length:[* TO 200]) AND (ec:3.2.1.17) AND (xref:cazy-GH22)`

- **$\alpha$-lactalbumin**: `(protein_name:"alpha lactalbumin") AND (fragment:false) NOT (existence:4) NOT (existence:5) AND (length:[* TO 200])`

- **myoglobin**: `(protein_name:"myoglobin") AND (xref:interpro-IPR002335) AND (fragment:false) NOT (existence:5) NOT (existence:4)`

- **hemoglobin-$\alpha$**: `(protein_name:"hemoglobin alpha") AND (xref:interpro-IPR002338) AND (fragment:false) NOT (existence:5) NOT (existence:4)`

- **hemoglobin-$\beta$**: `(protein_name:"hemoglobin beta") AND (xref:interpro-IPR002337) AND (fragment:false) NOT (existence:5) NOT (existence:4)`

- **trypsin**: `(protein_name:trypsin) AND (fragment:false) AND (ec:3.4.21.4) NOT (existence:5)`

- **chymotrypsin**: `(protein_name:chymotrypsin) AND (fragment:false) AND (ec:3.4.21.1) NOT (existence:5)`

- **tubulin-$\alpha$**: `(protein_name:"tubulin alpha") AND (family:"tubulin family") AND (length:[300 TO 600]) AND (fragment:false) NOT (annotation_score:1) NOT (annotation_score:2)`

- **tubulin-$\beta$**: `(protein_name:"tubulin beta") AND (family:"tubulin family") AND (length:[300 TO 600]) AND (fragment:false) NOT (annotation_score:1) NOT (annotation_score:2)`

- **interleukin-1 $\alpha$** `(protein_name:"interleukin-1 alpha") AND (family:il-1) AND (fragment:false) NOT (existence:4) NOT (existence:5) AND (length:[200 TO 400]) NOT (annotation_score:1)`

- **interleukin-1 $\beta$**: `(protein_name:"interleukin-1 beta") AND (family:il-1) AND (fragment:false) NOT (existence:4) NOT (existence:5) AND (length:[200 TO 400]) NOT (annotation_score:1)`

- **Histone H2A**: `(protein_name:"histone h2a") AND (family:histone) AND (fragment:false) NOT (existence:4) NOT (existence:5) AND (length:[* TO 200])`

- **Histone H2B**: `(protein_name:"histone h2b") AND (family:histone) AND (fragment:false) NOT (existence:4) NOT (existence:5) AND (length:[* TO 200])`

- **Cytochrome P450 CYP3**: `(family:"Cytochrome P450") AND ((gene:cyp3) OR (gene:cyp3A*)) AND (fragment:false) NOT (existence:4) NOT (existence:5) NOT (annotation_score:1)`

- **Cytochrome P450 CYP51**: `(family:"Cytochrome P450") AND ((gene:cyp51) OR (gene:cyp51A*) OR (gene:cyp51B*) OR (gene:cyp51C*)) AND (fragment:false) NOT (existence:4) NOT (existence:5) NOT (annotation_score:1)`

The GPCR sequences were collected from the GPCR-PEn database (URL: `https://gpcr.utep.edu/`) (Begum et al., 2020). Sequence redundancy of the rhodopsin-like family was reduced using CD-hit (Fu et al., 2012) with 30% sequence similarity cutoff.

## A.2 Code

The code to reproduce the computational experiments is available at `https://anonymous.4open.science/r/AFS_AAC_SVM-F3D9`. Protein sequences used in the computational experiments along with their UniProt IDs, are provided in the `datasets` folder as `.csv` files for each family.

# B Sequence and function diversity of protein classes within a dataset

Paralogous proteins have a common ancestor but have diverged in functionality. Protein functions are an aggregate of descriptors describing protein's activity and influence at various levels. They can be at the molecular level, like binding with specific molecules and catalysing reactions, to the biological process level, like energy metabolism. In B.1, we discuss the diversity of the functions of the proteins considered in our datasets.

As paralogs have a common ancestor, high sequence similarity would suggest high evolutionary conservation in the proteins. In B.2, we discuss the extent of sequence diversity in protein classes considered in our datasets.

We see that the dataset of proteins considered in our computational experiments are diverse in their function and sequences.

## B.1 Function diversity

We have considered paralogous proteins with varying functional differences. We find very subtle differences in the functions of trypsin and chymotrypsin. On the other hand, the function difference is drastic in the case of alpha-lactalbumin and lysozyme c.

Trypsin and chymotrypsin are a family of enzymes that break peptide bonds in proteins. The difference in the function of these proteins is fine-grained; trypsins cleave only the peptide bond following a basic amino acid ($K$ and $R$), while chymotrypsins cleave the peptide bond following a hydrophobic amino acid ($F$, $W$, and $Y$) (Dodson & Wlodawer, 1998).

GPCRs constitute a large and diverse class of cell surface receptor proteins. They trigger intra-cellular pathways in response to external signals. These signals are in the form of small molecules, called ligands. Depending upon the nature of ligands and other 3D structural similarities, GPCRs are grouped into distinct classes. We consider three such classes viz., rhodopsin-like, secretin-like, and glutamate-like. Further, we consider pairwise three subfamilies of rhodopsin-like GPCRs viz., aminergic receptors, lipid receptors, and peptide receptors.

Lysozyme C and $\alpha$-lactalbumin are sequence and structure homologs with mutually exclusive functions and high fold conservation. Based on phylogenetic analysis, they are considered to have diverged from a common ancestor millions of years ago (Qasba et al., 1997).

Globins are a superfamily of functionally divergent homologous protein families with a high level of fold conservation. We consider three well-known globin families viz., myoglobin, hemoglobin-$\alpha$ and hemoglobin-$\beta$. Myoglobin is a monomer that binds and releases oxygen as per physiological requirements. On the other hand, $\alpha$ and $\beta$ chains together constitute hemoglobin, a tetramer of composition $\alpha_2\beta_2$ (Dill et al., 2017), that transports oxygen in red blood cells.

Tubulin-$\alpha$ and tubulin-$\beta$ are similar to the hemoglobin-$\alpha$ and hemoglobin-$\beta$ pair in that they both share sequence and 3D structural similarities but have subtle functional differences. One copy each of tubulin-$\alpha$ and tubulin-$\beta$ form a functional dimer. Notably, neither two copies of tubulin-$\alpha$ nor two copies of tubulin-$\beta$ can form a functional dimer. Tubulin-$\beta$ has a catalytic activity (GTP hydrolysis) that is absent in tubulin-$\alpha$. This is one of the several subtle functional differences between tubulin-$\alpha$ and tubulin-$\beta$.

Interleukin-1 alpha and interleukin-1 beta are both proteins involved in the immune system. They differ from each other in their occurrence within the body (on cell surface or in blood circulation), activation mechanisms, and associated signalling pathways (Galozzi et al., 2021).

Cytochrome P450 (abbreviated as CYP) is a family of proteins whose function is clearance of 'foreign' molecules (drugs; also called as xenobiotics) as well as in certain biosynthesis pathways e.g., of steroid hormones. CYP3 and CYP51 are two of the several classes of CYPs; CYP3 metabolizes lipophilic molecules (McArthur et al., 2003) whereas CYP51 is involved in steroid biosynthesis (Hargrove et al., 2012).

Hemoglobin-$\alpha$/hemoglobin-$\beta$, histone H2A / histone H2B and tubulin-$\alpha$/tubulin-$\beta$ are paralog pairs that together function as heteromers (protein complexes made up of different protein subunits).

### B.2 Sequence Diversity

The dataset of the 15 paralog pairs in our experiments comprises 21 protein families (Table 2). For these families, we compute the within-class sequence similarities (for sequences within a protein family). We also compute the inter-class sequence similarities (between sequences from two different protein families) for each paralog pair. These are shown in Appendix Figure 5. We use a longest subsequence based similarity score, *lcss*, that is defined in B.2.1. In B.2.2, we see that *lcss* significantly varies across the 21 protein families we are considering as compared to its variation between the two protein sequences of any paralog pair.

#### B.2.1 Longest common subsequence based similarity score (*lcss*)

We compute the longest common subsequence (*lcs*) based similarity score (*lcss*) between a pair of protein sequences. We define *lcss* between two sequences as the length of their longest common subsequence, *lcs*, divided by the length of the longest sequence from the two. For a pair of protein sequences, $\mathbf{p}^{(i)} = (p_1^{(i)}, p_2^{(i)}, \ldots, p_{L_1}^{(i)})$ of length $L_1$ and $\mathbf{p}^{(j)} = (p_1^{(j)}, p_2^{(j)}, \ldots, p_{L_2}^{(j)})$ of length $L_2$, their *lcss* is,

$$lcs(\mathbf{p}^{(i)}, \mathbf{p}^{(j)}) = \max_{\mathbf{q}} k$$

$$\text{s.t. } \mathbf{q} = (q_1, q_2, \ldots, q_k)$$
$$(q_1 = p_{x_1}^{(i)} = p_{y_1}^{(j)}, q_2 = p_{x_2}^{(i)} = p_{y_2}^{(j)}, \ldots, q_k = p_{x_k}^{(i)} = p_{y_k}^{(j)})$$
$$x_1 < x_2 < \ldots < x_k$$
$$y_1 < y_2 < \ldots < y_k$$

*lcs* based similarity score, *lcss*, is defined as,

$$lcss(\mathbf{p}^{(i)}, \mathbf{p}^{(j)}) = \frac{lcs(\mathbf{p}^{(i)}, \mathbf{p}^{(j)})}{\max(L_1, L_2)} \in [0, 1]$$

$lcss(\mathbf{p}^{(i)}, \mathbf{p}^{(j)}) = 1$ if and only if $\mathbf{p}^{(i)} = \mathbf{p}^{(j)}$, i.e., sequences are identical. Whereas $lcss(\mathbf{p}^{(i)}, \mathbf{p}^{(j)}) = 0$ if and only if $p_x^{(i)} \neq p_y^{(j)}, \forall x, y$, i.e., there are no amino acids common to both the sequences.

#### B.2.2 Within-class and inter-class *lcss* for the 15 paralog pairs

**Within-class** *lcss*: $lcss(\mathbf{p}^{(i)}, \mathbf{p}^{(j)})$ are computed with $\mathbf{p}^{(i)}, \mathbf{p}^{(j)}$ from the same protein family. These are shown in *blue* and *magenta* in Figure 5 (with box-plots) for each of 21 protein families in the 15 paralog pairs.

- 12 of 21 protein families have median within-class *lcss* greater than 0.5. This implies less sequence diversity in this set of families from the remaining families. These are,
    - Median *lcss* $\geqslant$ 0.6 for 11 of these 12 families and $\geqslant$ 0.8 for 3 families (high level of sequence conservation).
    - For 7 out of the 15 paralog pairs, the median within-class *lcss* > 0.5 for both families of a paralogous pair.
- For the remaining 9 protein families, the median within-class *lcss* is less than 0.5. This implies high sequence diversity in this set of families from the remaining families. These are,

Table 3: The median within-class *lcss* between sequences from the respective families. See boxplot in Figure 5.

| Family | $\alpha$-lactalbumin | lysozyme C | myoglobin | hemoglobin-$\alpha$ | hemoglobin-$\beta$ |
|---|---|---|---|---|---|
| Median *lcss* | 0.6 | 0.59 | 0.81 | 0.63 | 0.67 |

| Family | tubulin-$\alpha$ | tubulin-$\beta$ | cytochrome P450 CYP3 |
|---|---|---|---|
| Median *lcss* | 0.83 | 0.82 | 0.7 |

| Family | interleukin-1 $\alpha$ | interleukin-1 $\beta$ | histone H2A | histone H2B |
|---|---|---|---|---|
| Median *lcss* | 0.72 | 0.66 | 0.65 | 0.68 |

Table 4: The median within-class *lcss* between sequences from the respective families. See boxplot in Figure 5.

| Family | cytochrome P450 CYP51 | rhodopsin-like receptor | aminergic receptor | lipid receptor |
|---|---|---|---|---|
| Median *lcss* | 0.47 | 0.34 | 0.39 | 0.37 |

| Family | peptide receptor | glutamate-like receptor | secretin-like receptor | trypsin | chymotrypsin |
|---|---|---|---|---|---|
| Median *lcss* | 0.37 | 0.35 | 0.36 | 0.47 | 0.45 |

- For 7 out of the 15 paralog pairs, the median within-class $lcss < 0.5$ for both families of a paralogous pair.

- For the paralog pair Cytochrome P450 CYP3 vs CYP51, the median sequence similarity for CYP3 is greater than 0.5, while for CYP51, it is less than 0.5.

**Inter-class** *lcss*: $lcss(\mathbf{p}^{(i)}, \mathbf{p}^{(j)})$ are computed with $\mathbf{p}^{(i)}, \mathbf{p}^{(j)}$ respectively from two protein families that are paralog pairs. These are shown in *cyan* in Figure 5 (with box-plots) for each of the 15 paralog pairs.

- The median inter-class *lcss* is less than 0.5 for all paralog pairs. This implies sequences of the proteins across the classes are not very similar.

**Distinguishing paralog pairs based on within-class and inter-class** *lcss*: If we analyse the box plots in Figure 5 - two paralog pair proteins can be considered to be distinguishable based on sequence similarity if the upper-whisker of inter-class *lcss* is lower than the lower-whiskers of the respective within-class *lcss* scores.

- Apart from paralog pairs, tubulin-$\alpha$ vs tubulin-$\beta$ (Figure 5c) and interleukin-1 $\alpha$ vs interleukin-1 $\beta$ (Figure 5d), no other paralog pair is distinguishable based on sequence similarity.

- For Trypsin vs Chymotrypsin and the 6 GPCR pairs (Figures 5b and 5j to 5o), the median inter-class *lcss* scores are close to the within-class *lcss* scores making them indistinguishable based on sequence similarity.

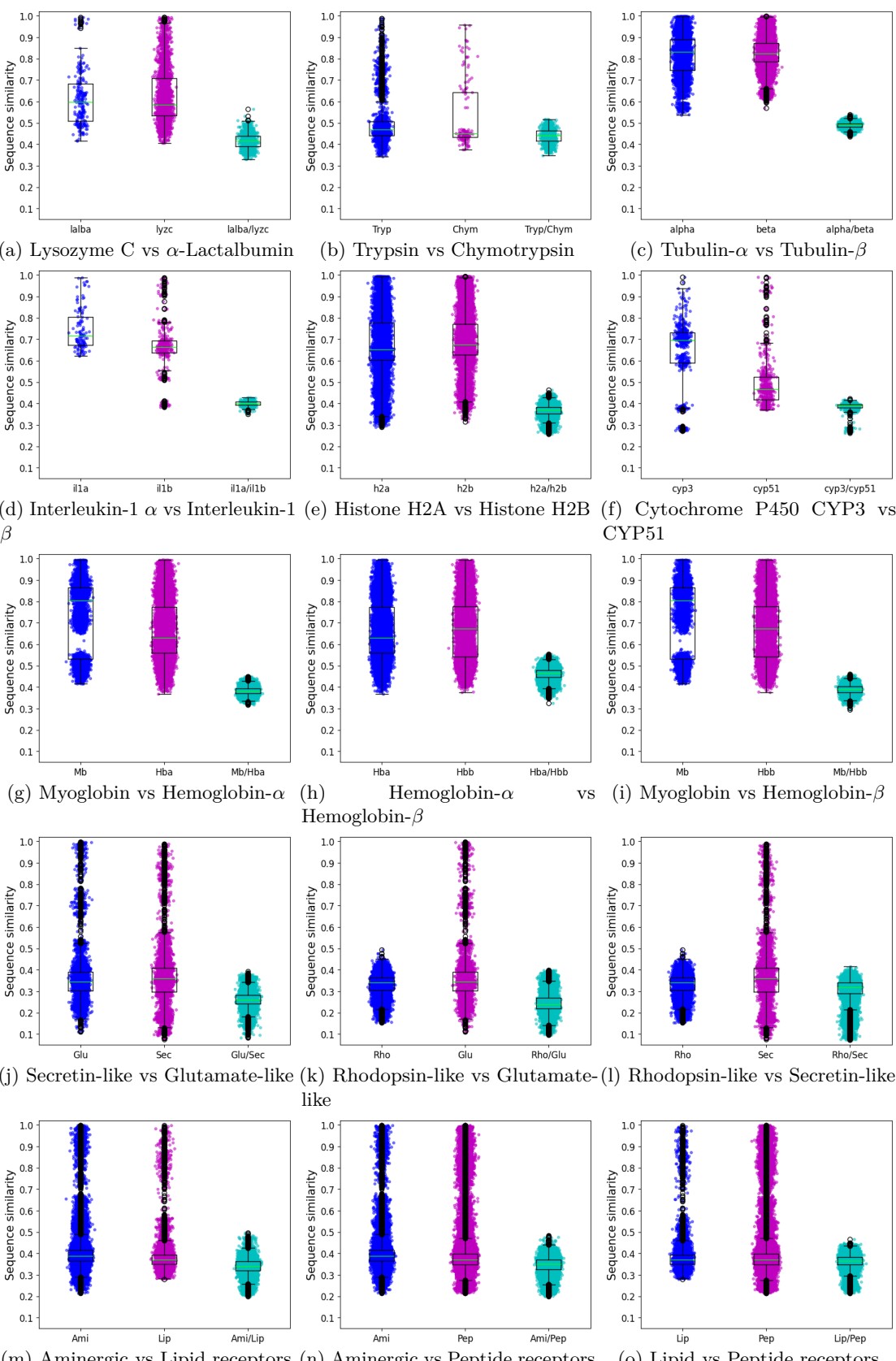

Figure 5: *lcss* sequence similarity scores for the 15 paralog pair datasets. In the boxplots, the lower and upper whiskers are at 1.5 IQR (inter-quantile range) values away from the first and third quartiles respectively.

## C   The SVEA algorithm for AFS

---

**Algorithm 1** $\phi_i$ Monte-carlo approximation algorithm as suggested in (Tripathi et al., 2020; 2021)

---

**Input:** Feature set $N = \{1, 2, \ldots, 20\}$, Number of sample permutations $samPerm$, Datasets $(D_P, D_Q)$, Set of coalitions $Sam\_co\_set = [()]$
**Initialise:** $v(()) = 0$, $\hat{\phi}_i := 0 \forall i \in N$
Append $N$ to $Sam\_co\_set$.
**for** $s = 1, 2, \ldots, samPerm$ **do**
  Take $\pi \in PermSet(N)$ with probability $\frac{1}{20!}$.
  **for** $i = 1, 2, \ldots, 20$ **do**
    Compute $Pred^i(\pi) = \{\pi(1), \pi(2), \ldots \pi(k-1) | i = \pi(k)\}$
    **if** $Pred^i(\pi)$ not in $Sam\_co\_set$ **then**
      Compute $v(Pred^i(\pi)) = 1 - tr\_er(Pred^i(\pi))$.
      Append $Pred^i(\pi)$ to $Sam\_co\_set$.
    **end if**
    **if** $Pred^i(\pi) \cup i$ not in $Sam\_co\_set$ **then**
      Compute $v(Pred^i(\pi) \cup \{i\}) = 1 - tr\_er(Pred^i(\pi) \cup \{i\})$.
      Append $Pred^i(\pi) \cup \{i\}$ to $Sam\_co\_set$.
    **end if**
    $\hat{\phi}_i = \hat{\phi}_i + v(Pred^i(\pi) \cup \{i\}) - v(Pred^i(\pi)))$
  **end for**
**end for**
$\hat{\phi}_i = \frac{\hat{\phi}_i}{samPerm}, \forall i \in N$

---

## D   SVM training for $AFS$ partition

We provide details for the linear SVM classifier discussed in Section 2.3. We use 5-fold cross-validation to tune the SVM regularisation hyperparameter $C$ from $\{0.1, 1, 10, 100, 1000\}$ that gives the best average classification score for the 5 folds. $C$ is inversely proportional to the strength of regularisation. In general, we find that there is an imbalance in the number of sequences that we find for the two paralogous proteins, i.e. say $n_P >> n_Q$. It is known that accuracy is not a well-suited performance measure of the classifier in class imbalance settings. Therefore, we use the arithmetic mean of sensitivity and specificity (AM) to measure the performance of the classifier (Brodersen et al., 2010). Further, we use a class-balanced version of hinge loss for training the SVM as suggested in (Menon et al., 2013) for statistical consistency with the AM score. Appendix Table 5 reports the train and test scores of the trained linear SVM with AAC and $AFS$ features, respectively, on the protein family datasets (See Appendix Table 2) considered in our computational experiments.

## E  More details for computational experiments

We present here details on some of the computational experiments that have been referenced in the main paper.

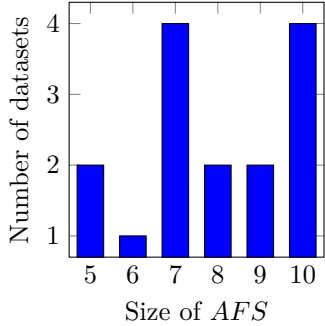

Figure 6: The sizes of the $AFS$ for the 15 datasets.

(a) Lysozyme C vs $\alpha$-Lactalbumin

(b) Trypsin vs Chymotrypsin

(c) Tubulin-$\alpha$ vs Tubulin-$\beta$

(d) Interleukin-1 $\alpha$ vs Interleukin-1 $\beta$

(e) Histone H2A vs Histone H2B

(f) Cytochrome P450 CYP3 vs CYP51

(g) Aminergic vs Lipid receptors

(h) Aminergic vs Peptide receptors

(i) Lipid vs Peptide receptors

Figure 7: (Continued from Figure 3) Shapley value ($\phi(i)$) for AAC features computed using SVEA.

Table 5: Classification scores for different pairs of paralogous proteins using the train/test datasets described in Table 2, using AAC and $AFS$ features. The $AFS$ amino acids computed for each pair are given in Table 1. The train score is the mean ($\pm 1$ standard deviation) 5-fold cross-validation score. **AM** is the arithmetic mean of specificity and sensitivity. **Acc** is the accuracy.

(a) Lysozyme C vs $\alpha$-Lactalbumin

|  | AAC | AFS |
|---|---|---|
| **Train AM** | 1.0 | 0.993 ($\pm$0.013) |
| **Test AM** | 0.896 | 0.898 |
| **Train Acc** | 1.0 | 0.99 ($\pm$0.02) |
| **Test Acc** | 0.836 | 0.881 |

(b) Trypsin vs chymotrypsin

|  | AAC | AFS |
|---|---|---|
| **Train AM** | 0.992 ($\pm$0.015) | 0.977 ($\pm$0.031) |
| **Test AM** | 0.873 | 0.835 |
| **Train Acc** | 0.988 ($\pm$0.024) | 0.965 ($\pm$0.047) |
| **Test Acc** | 0.844 | 0.756 |

(c) Tubulin-$\alpha$ vs Tubulin-$\beta$

|  | AAC | AFS |
|---|---|---|
| **Train AM** | 0.996 ($\pm$0.009) | 0.997 ($\pm$0.006) |
| **Test AM** | 0.992 | 0.992 |
| **Train Acc** | 0.997 ($\pm$0.006) | 0.994 ($\pm$0.008) |
| **Test Acc** | 0.991 | 0.994 |

(d) Histone H2A vs Histone H2B

|  | AAC | AFS |
|---|---|---|
| **Train AM** | 0.983 ($\pm$0.016) | 0.983 ($\pm$0.01) |
| **Test AM** | 0.91 | 0.934 |
| **Train Acc** | 0.983 ($\pm$0.016) | 0.983 ($\pm$0.01) |
| **Test Acc** | 0.889 | 0.922 |

(e) Globins

| Dataset |  | AAC | AFS |
|---|---|---|---|
| Myoglobin vs Hemoglobin-$\alpha$ | **Train AM** | 0.998 ($\pm$0.003) | 0.994 ($\pm$0.009) |
|  | **Test AM** | 0.968 | 0.97 |
|  | **Train Acc** | 0.998 ($\pm$0.005) | 0.995 ($\pm$0.006) |
|  | **Test Acc** | 0.969 | 0.971 |
| Myoglobin vs Hemoglobin-$\beta$ | **Train AM** | 1.0 ($\pm$0.0) | 1.0 ($\pm$0.0) |
|  | **Test AM** | 0.957 | 0.936 |
|  | **Train Acc** | 1.0 ($\pm$0.0) | 1.0 ($\pm$0.0) |
|  | **Test Acc** | 0.949 | 0.919 |
| Hemoglobin-$\alpha$ vs Hemoglobin-$\beta$ | **Train AM** | 0.983 ($\pm$0.008) | 0.976 ($\pm$0.007) |
|  | **Test AM** | 0.961 | 0.935 |
|  | **Train Acc** | 0.983 ($\pm$0.008) | 0.976 ($\pm$0.006) |
|  | **Test Acc** | 0.966 | 0.947 |

(f) GPCRs

| Dataset |  | AAC | AFS |
|---|---|---|---|
| Secretin-like vs Glutamate-like | **Train AM** | 0.933 ($\pm$0.042) | 0.95 ($\pm$0.032) |
|  | **Test AM** | 0.888 | 0.845 |
|  | **Train Acc** | 0.933 ($\pm$0.042) | 0.95 ($\pm$0.032) |
|  | **Test Acc** | 0.889 | 0.844 |
| Rhodopsin-like vs Glutamate-like | **Train AM** | 0.884 ($\pm$0.042) | 0.85 ($\pm$0.045) |
|  | **Test AM** | 0.967 | 0.934 |
|  | **Train Acc** | 0.867 ($\pm$0.038) | 0.837 ($\pm$0.032) |
|  | **Test Acc** | 0.956 | 0.926 |
| Rhodopsin-like vs Secretin-like | **Train AM** | 0.917 ($\pm$0.051) | 0.878 ($\pm$0.065) |
|  | **Test AM** | 0.934 | 0.846 |
|  | **Train Acc** | 0.908 ($\pm$0.06) | 0.863 ($\pm$0.073) |
|  | **Test Acc** | 0.941 | 0.853 |
| Aminergic vs Lipid receptors | **Train AM** | 0.949 ($\pm$0.014) | 0.943 ($\pm$0.005) |
|  | **Test AM** | 0.922 | 0.843 |
|  | **Train Acc** | 0.943 ($\pm$0.017) | 0.94 ($\pm$0.008) |
|  | **Test Acc** | 0.92 | 0.84 |
| Aminergic vs Peptide receptors | **Train AM** | 0.835 ($\pm$0.06) | 0.818 ($\pm$0.053) |
|  | **Test AM** | 0.844 | 0.79 |
|  | **Train Acc** | 0.83 ($\pm$0.06) | 0.819 ($\pm$0.051) |
|  | **Test Acc** | 0.827 | 0.784 |
| Lipid vs Peptide receptors | **Train AM** | 0.829 ($\pm$0.022) | 0.76 ($\pm$0.035) |
|  | **Test AM** | 0.845 | 0.709 |
|  | **Train Acc** | 0.838 ($\pm$0.018) | 0.75 ($\pm$0.032) |
|  | **Test Acc** | 0.858 | 0.725 |

(g) Interleukin-1 $\alpha$ vs Interleukin-1 $\beta$

|  | AAC | AFS |
|---|---|---|
| **Train AM** | 0.98 ($\pm$0.04) | 0.98 ($\pm$0.04) |
| **Test AM** | 0.979 | 0.985 |
| **Train Acc** | 0.975 ($\pm$0.05) | 0.975 ($\pm$0.05) |
| **Test Acc** | 0.961 | 0.971 |

(h) Cytochrome P450 CYP3 vs Cytochrome P450 CYP51

|  | AAC | AFS |
|---|---|---|
| **Train AM** | 0.967 ($\pm$0.041) | 0.933 ($\pm$0.062) |
| **Test AM** | 0.902 | 0.92 |
| **Train Acc** | 0.969 ($\pm$0.038) | 0.936 ($\pm$0.062) |
| **Test Acc** | 0.894 | 0.908 |

### E.1 Tubulin

The inter-chain contact residues from the tubulin-$\alpha/\beta$ heterodimer were identified using ChimeraX 1.4 (Pettersen et al., 2021). The *Contacts* tool available in *Tools → Structure Analysis* was used with settings as shown in Figure 8. For PDB ID:3JAR we count the residues of chain-A (tubulin-$\alpha$) and chain-B (tubulin-$\beta$) which are in contact with the residues of other tubulin chains. Similarly, for PDB ID:5N5N we count the residues of chain-G (tubulin-$\alpha$) and chain-B (tubulin-$\beta$) which are in contact with the residues of other tubulin chains. The code for counting the $AFS$ residues at the identified contact points of the respective chains is available at `https://anonymous.4open.science/r/AFS_AAC_SVM-F3D9`.

### E.2 Histone

The inter-chain contact residues of histone H2A and H2B were identified from its heterooctameric structure comprising of two H2A/H2B dimers and one H3/H4 tetramer, using ChimeraX 1.4. The *Contacts* tool available in *Tools → Structure Analysis* was used with settings as shown in Figure 8. For PDB ID: 1AOI and 3KWQ, we count the residues of an H2A and an H2B chain, which are in contact with other histone chains in the heterooctameric structure. The code for counting the $AFS$ residues at the identified contact points of the respective chains is available at `https://anonymous.4open.science/r/AFS_AAC_SVM-F3D9`.

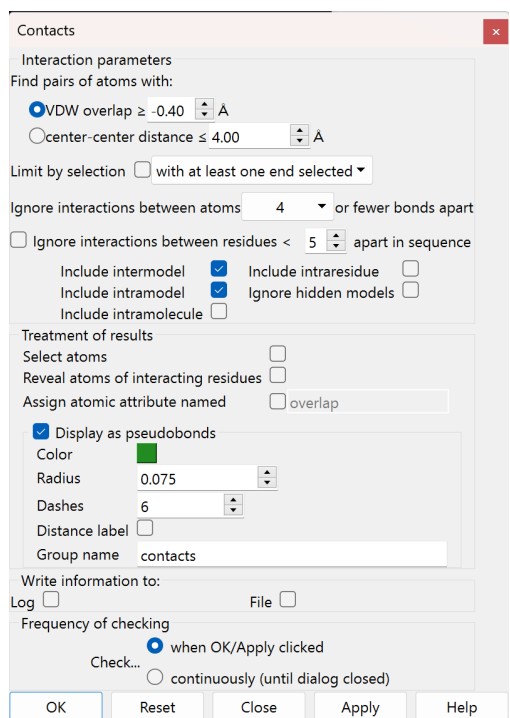

Figure 8: ChimeraX 1.4 settings for identifying inter-chain contact points from the tubulin-$\alpha/\beta$ heterodimer and from the histone heterooctamer

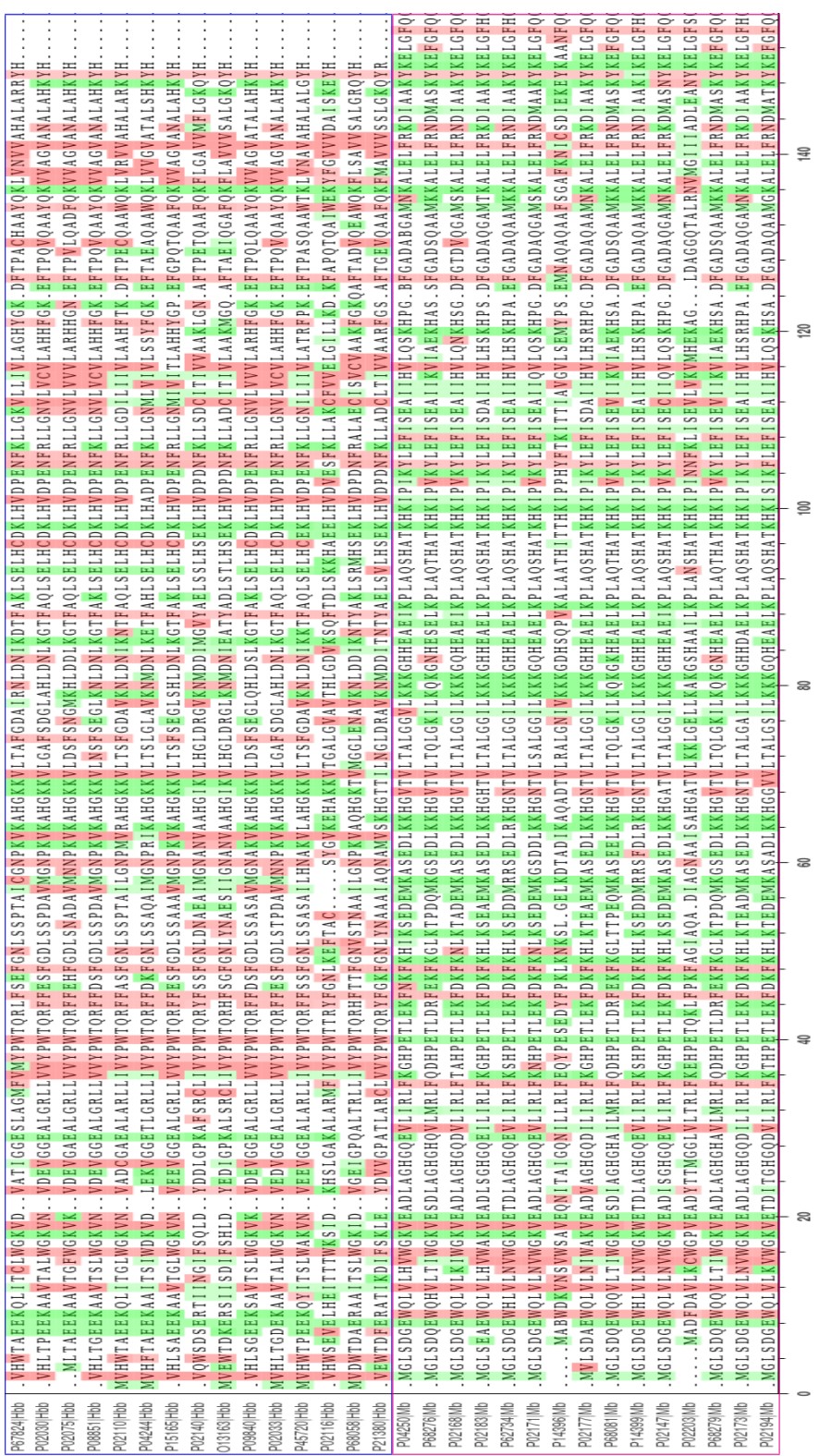

Figure 9: Multiple sequence alignment of hemoglobin-$\beta$ and myoglobin sequences. 15 sequences on the left are from hemoglobin-$\beta$ and on the right are from myoglobin. The sequences are randomly selected from the train set of the protein families. $AFS$(Myoglobin) amino acids are in green and $AFS$(Hemoglobin-$\beta$) in red. The intensity of the color is proportional to the Shapley value $\phi(i)$ of the amino acid $i$ (See Figure 3c)

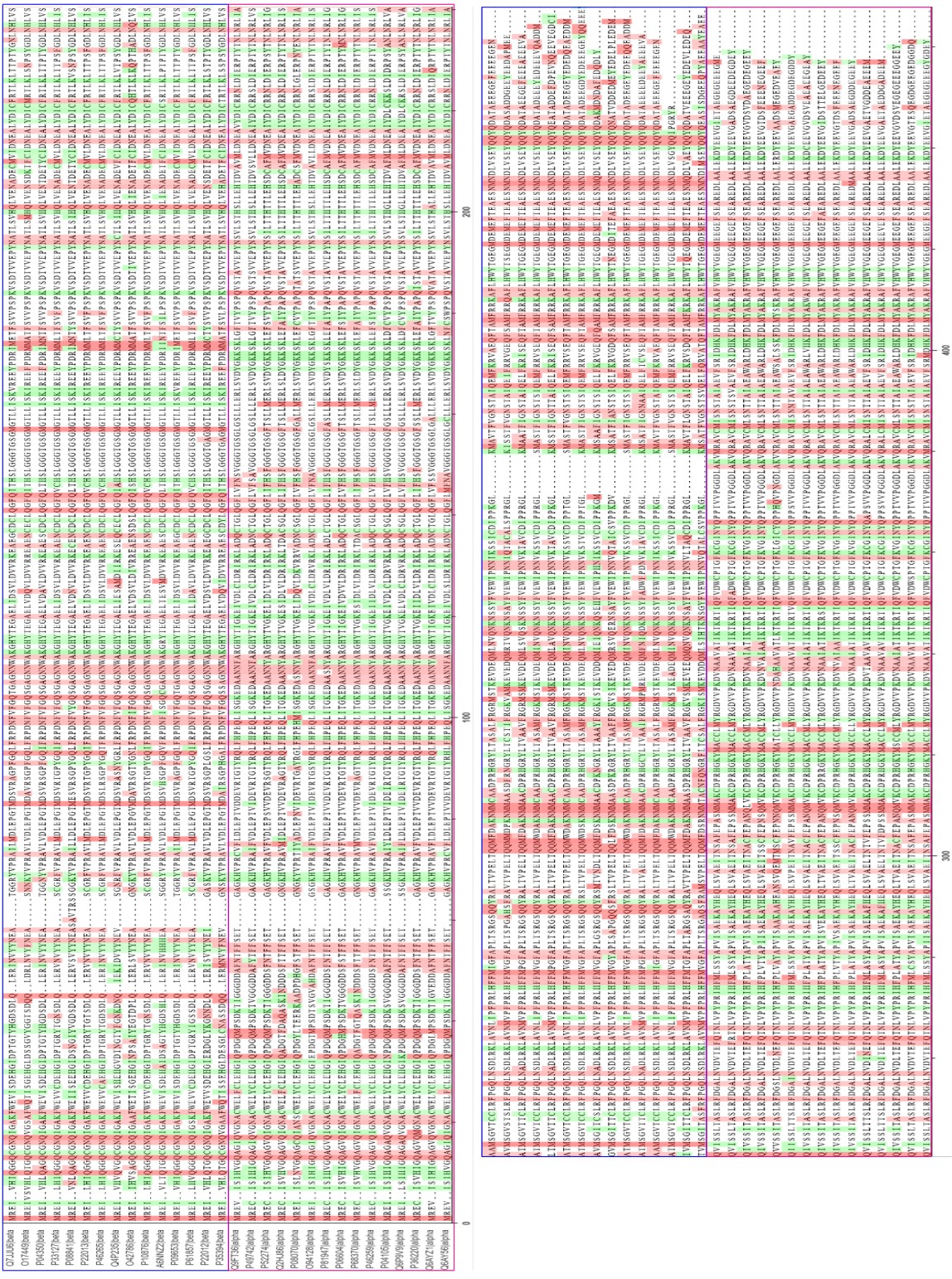

Figure 10: Multiple sequence alignment of tubulin-$\alpha$ and tubulin-$\beta$ sequences. 15 sequences on the left are from tubulin-$\beta$ and on the right are from tubulin-$\alpha$. The sequences are randomly selected from the train set of the protein families. $AFS$(Tubulin-$\alpha$) amino acids are in green and $AFS$(Tubulin-$\beta$) in red. The intensity of the color is proportional to the Shapley value $\phi(i)$ of the amino acid $i$ (See Figure 7c)

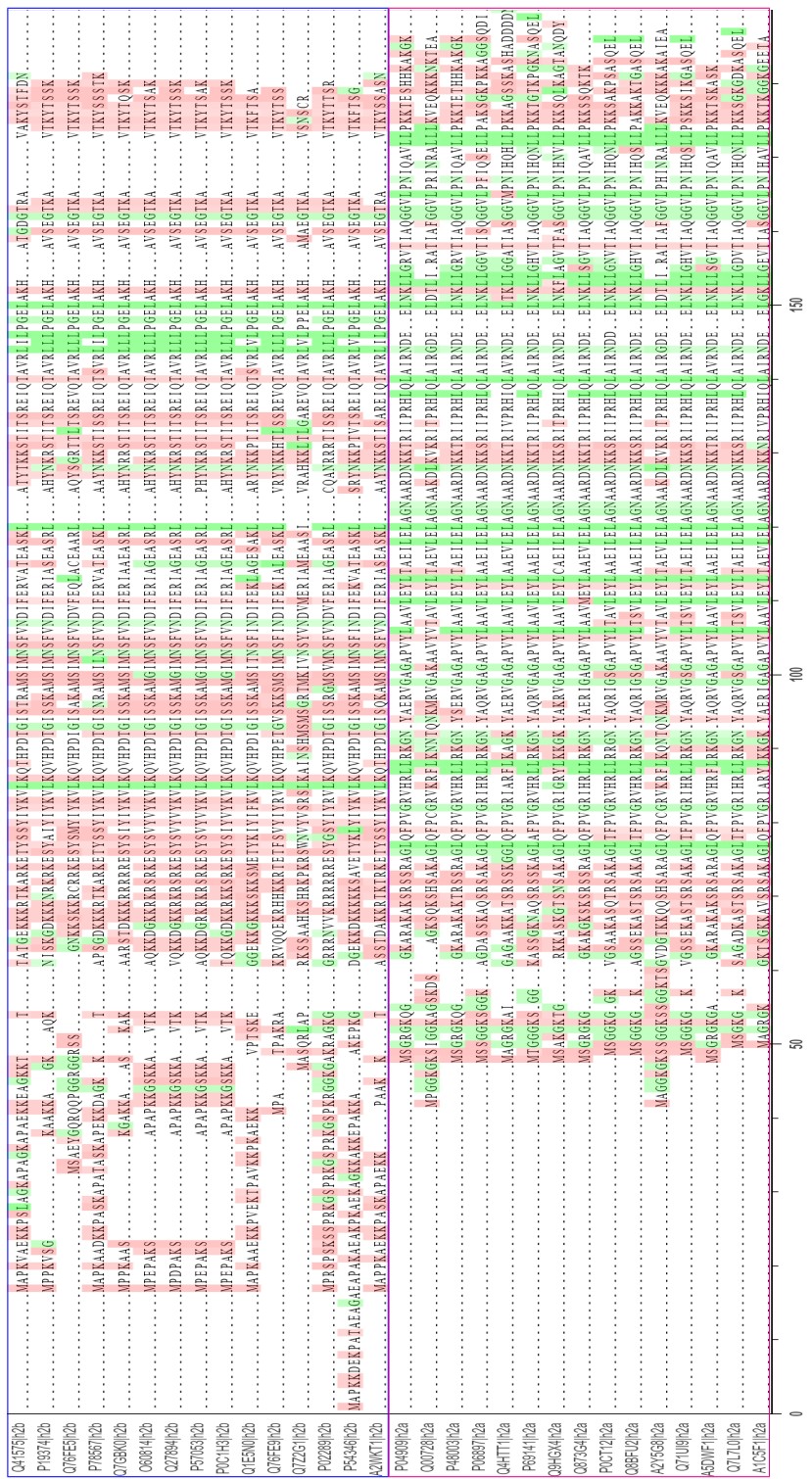

Figure 11: Multiple sequence alignment of histone H2A and histone H2B sequences. 15 sequences on the left are from histone H2B and on the right are from histone H2B. The sequences are randomly selected from the train set of the protein families. $AFS$(Histone H2A) amino acids are in green and $AFS$(Histone H2B) in red. The intensity of the color is proportional to the Shapley value $\phi(i)$ of the amino acid $i$ (See Figure 7e)

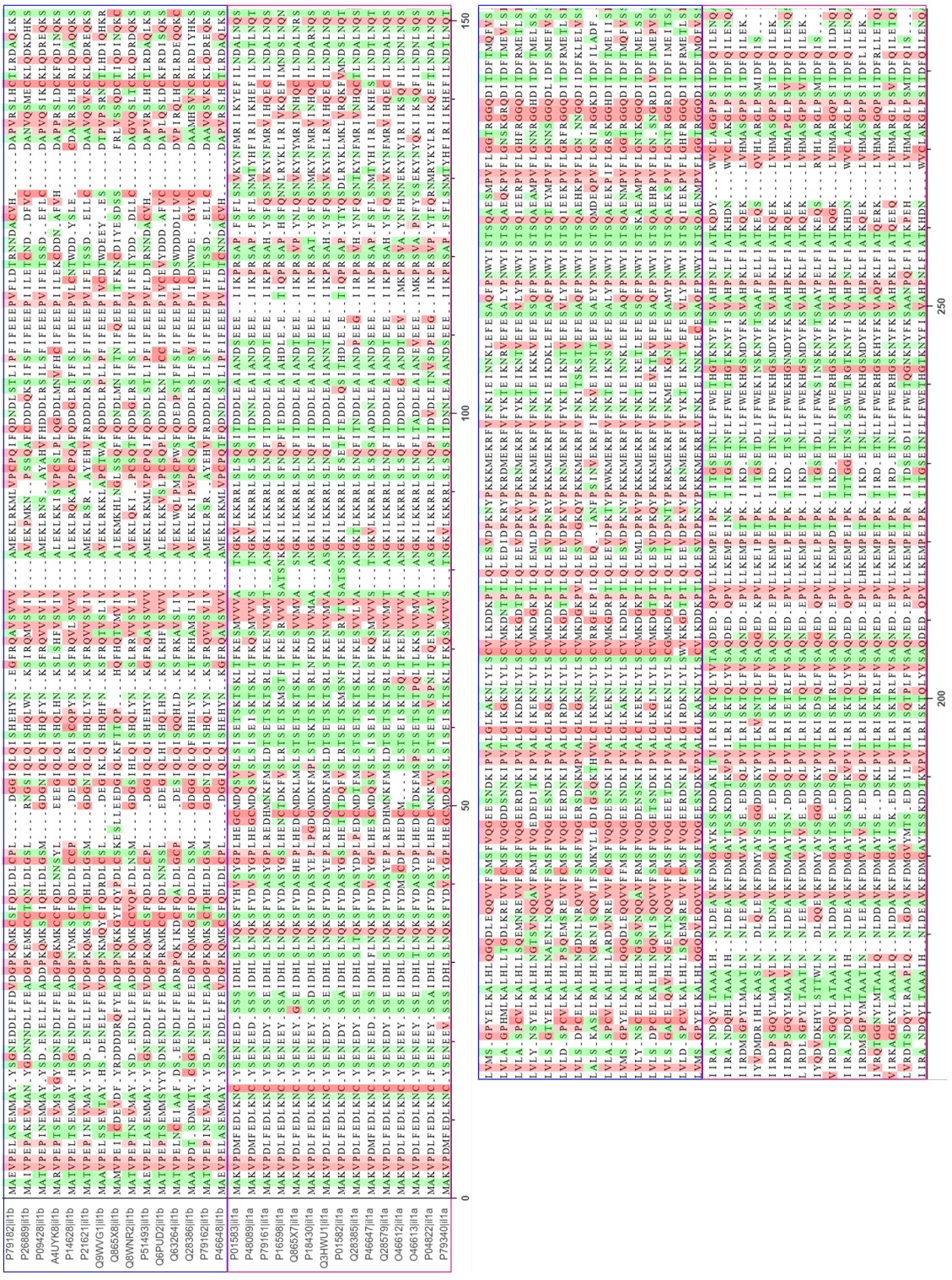

Figure 12: Multiple sequence alignment of interleukin-1 α and interleukin-1 β sequences. 15 sequences on the left are from interleukin-1 β and on the right are from interleukin-1 α. The sequences are randomly selected from the train set of the protein families. $AFS$(Interleukin-1 α) amino acids are in green and $AFS$(Interleukin-1 β) in red. The intensity of the color is proportional to the Shapley value $\phi(i)$ of the amino acid $i$ (See Figure 7d)

### E.3 Marginal contribution feature importance (MCI) (Catav et al., 2021) for $AFS$

We validate the top-selected features, $AFS$, in our experiments using the MCI feature rankings (Catav et al., 2021). The MCI method, by design, was aimed at addressing the issue of correlated features. Catav et al. (2021), show that Shapley value-based feature rankings are altered when correlated features are introduced. Therefore, they propose the MCI method using a different set of axioms to circumvent the issue of correlated features. Thus, we validate the Shapley value-based feature rankings with those by MCI. We find strong agreement in the two rankings (see Table 6), suggesting a lack of correlated features for the datasets considered in our experiments. We use Shapley value instead of MCI in our main pipeline for computing $AFS$, as the efficiency axiom of the Shapley value allows us to draw a fair cutoff line for selecting the important features. Since MCI doesnt satisfy the efficiency axiom, the same cutoff line may not be applicable to it.

For a feature $i$, its MCI score is defined as,

$$MCI(i) = \max_{S \subseteq N \setminus \{i\}} v(S \cup \{i\}) - v(S),$$

Here, $v(\cdot)$ is the same as that defined in Section 2.2. We compare the amino acids with the top-$d$ ($d$ = size of $AFS$) MCI scores to the $AFS$ in Table 6. MCI is computed using the same approximation scheme as in Appendix Section C Algorithm 1 with appropriate modifications.

Table 6: $AFS$ comparison with the amino acids having the top-$d$ MCI (Catav et al., 2021) scores. Here, $d$ is the size of $AFS$ for the respective dataset. The amino acids that differ in the two sets are in **bold and underlined**, with their counts mentioned in the rightmost column. For 8 of 15 datasets, $AFS$ and top-$d$ MCI sets are the same, while only for two datasets do they differ in two amino acids. For all 15 datasets, at least the top-3 MCI amino acids are in $AFS$. For 11 of these datasets, at least the top-5 MCI amino acids are in $AFS$.

| Paralog pair | top-$d$ MCI amino acids (rank-1 → rank-$d$) | $AFS$ | Difference count |
|---|---|---|---|
| Lysozyme C (74) and $\alpha$-Lactalbumin (22) | $\{I, A, D, G, R, F, N, E, W, L\}$ | $\{I, A, D, N, G, R, E, F, L, W\}$ | 0 |
| Trypsin (66) and Chymotrypsin (17) | $\{Y, W, T, A, K, V, \underline{\boldsymbol{I}}\}$ | $\{Y, W, T, A, V, K, \underline{\boldsymbol{P}}\}$ | 1 |
| Tubulin-$\alpha$ (117) and Tubulin-$\beta$ (191) | $\{Q, M, K, H, F, I, N, A, Y, C\}$ | $\{M, Q, K, N, F, I, H, A, C, Y\}$ | 0 |
| Histone H2A (180) and Histone H2B (177) | $\{L, G, K, S, M, T, N, F, Y\}$ | $\{L, G, S, M, K, N, T, Y, F\}$ | 0 |
| Interleukin-1 $\alpha$ (16) and Interleukin-1 $\beta$ (25) | $\{G, C, T, V, Q, S, A, \underline{\boldsymbol{I}}, P\}$ | $\{C, G, T, S, V, Q, A, \underline{\boldsymbol{N}}, P\}$ | 1 |
| Cytochrome P450 CYP3 (32) and CYP51 (32) | $\{H, F, G, K, A, P, N\}$ | $\{H, F, G, K, A, P, N\}$ | 0 |
| **Globins** | | | |
| Myoglobin (107) and Hemoglobin-$\alpha$ (303) | $\{V, Y, E, K, S, G, W, I, C, P\}$ | $\{E, S, Y, V, K, P, I, G, C, W\}$ | 0 |
| Myoglobin (107) and Hemoglobin-$\beta$ (285) | $\{V, K, E, C, W, N, F, Y, M, I\}$ | $\{K, V, C, E, W, N, F, M, Y, I\}$ | 0 |
| Hemoglobin-$\alpha$ (303) and Hemoglobin-$\beta$ (285) | $\{W, S, N, P, \underline{\boldsymbol{V}}\}$ | $\{W, P, N, S, \underline{\boldsymbol{G}}\}$ | 1 |
| **GPCRs** | | | |
| Rhodopsin-like (181) and Glutamate-like (89) | $\{D, E, Q, G, L, \underline{\boldsymbol{I}}\}$ | $\{D, Q, E, G, \underline{\boldsymbol{M}}, L\}$ | 1 |
| Secretin-like (90) and Glutamate-like (89) | $\{W, H, Y, V, D\}$ | $\{W, H, Y, V, D\}$ | 0 |
| Rhodopsin-like (181) and Secretin-like (90) | $\{W, E, H, Q, S, M, V, A\}$ | $\{W, E, M, S, V, H, Q, A\}$ | 0 |
| **Rhodopsin-like GPCRs** | | | |
| Aminergic receptors (186) and Lipid receptors (113) | $\{L, E, P, \underline{\boldsymbol{K}}, F, D, \underline{\boldsymbol{I}}\}$ | $\{L, P, E, \underline{\boldsymbol{W}}, F, \underline{\boldsymbol{M}}, D\}$ | 2 |
| Aminergic receptors (186) and Peptide receptors (367) | $\{L, E, K, F, M, \underline{\boldsymbol{H}}, R, D\}$ | $\{L, F, E, M, K, D, \underline{\boldsymbol{V}}, R\}$ | 1 |
| Lipid receptors (113) and Peptide receptors (367) | $\{R, G, P, \underline{\boldsymbol{K}}, I, V, \underline{\boldsymbol{T}}\}$ | $\{P, R, G, I, \underline{\boldsymbol{W}}, \underline{\boldsymbol{S}}, V\}$ | 2 |

### E.4 Variance in Shapley value estimates and $AFS$

We computed the Shapley value estimates across 10 Monte Carlo runs with different random seeds. We do not find significant variation in these estmiates across the different runs as seen in Figure 13.

We also looked at the $AFS$ identified using the Shapley value in each run. We do not find substantial differences in the $AFS$. As a stronger check of consistency, for each paralog pair, we computed the intersection of the $AFS$s across the 10 runs, see Table 7. We find that for each paralog pair, this intersection of $AFS$s recovers at least 66.6% (two-thirds) of the amino acids from the (single-run) $AFS$ reported in the paper. For 14 pairs, greater than 75% is recovered. For 4 pairs, 100% is recovered. Furthermore, the amino acids with the top-3 Shapley values reported in the paper are present in the intersection of $AFS$s for all paralog pairs. For 13 pairs, the top-5 amino acids are in the intersection.

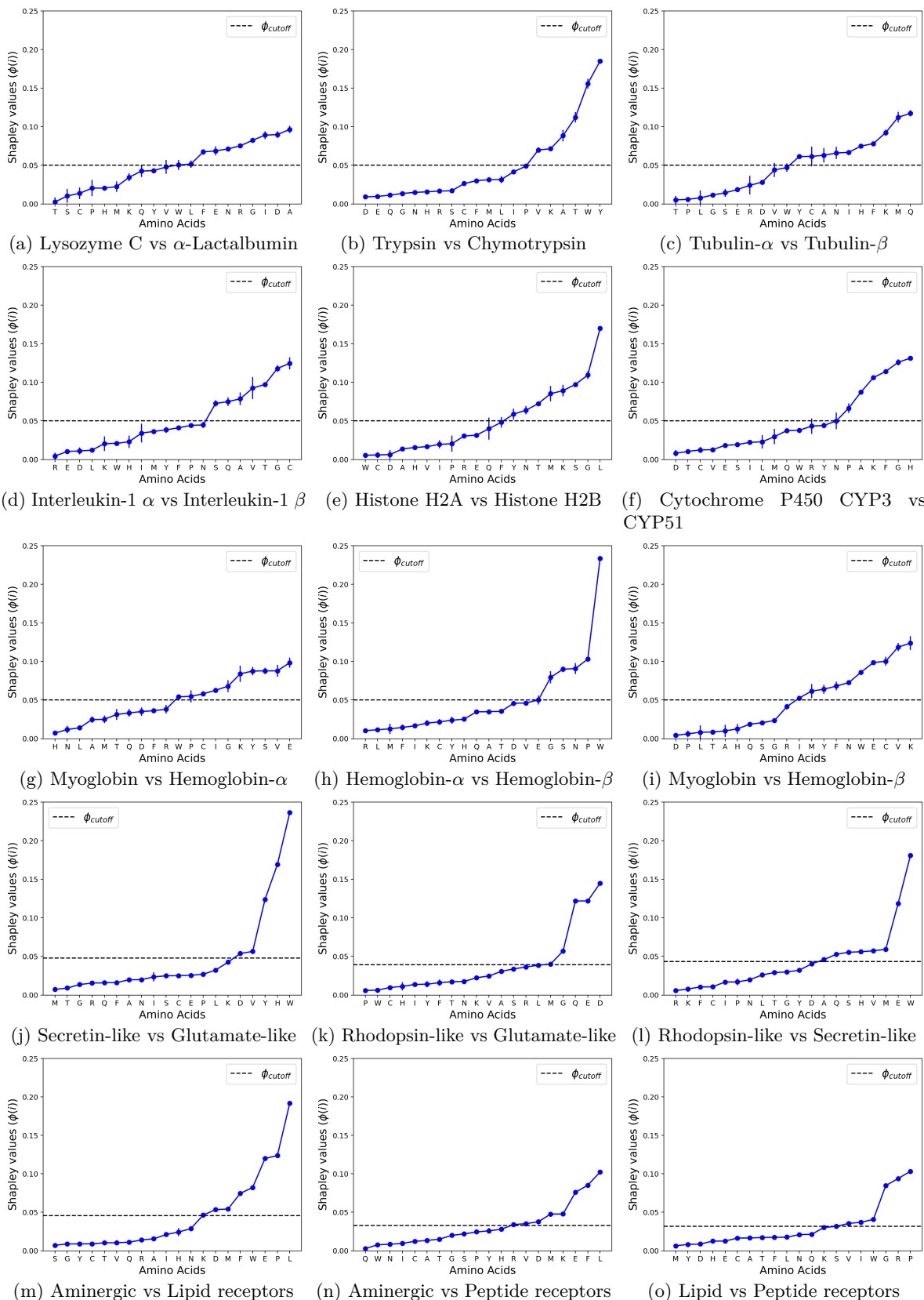

Figure 13: Mean (± standard deviation) of Shapley value estimates across 10 Monte Carlo runs with different random seeds. The cut-off $\phi_{cutoff}$ here is computed based on the mean Shapley values.

Table 7: Comparison of $AFS$ computed in 1 run with the intersection of $AFS$ computed over 10 runs. The amino acids that differ in the two sets are in **bold and underlined**. The percentage of overlap in the two sets is given in the rightmost column.

| Paralog pair | $AFS$ in 1 run | Intersection of $AFS$ over 10 runs | overlap ratio (%) |
|---|---|---|---|
| Lysozyme C and $\alpha$-Lactalbumin | $\{I, A, D, N, G, R, E, F, \underline{\textbf{L}}, \underline{\textbf{W}}\}$ | $\{I, A, D, N, G, R, E, F\}$ | 80 |
| Trypsin and Chymotrypsin | $\{Y, W, T, A, V, K, \underline{\textbf{P}}\}$ | $\{Y, W, T, A, V, K\}$ | 86 |
| Tubulin-$\alpha$ and Tubulin-$\beta$ | $\{M, Q, K, \underline{\textbf{N}}, F, I, H, A, C, \underline{\textbf{Y}}\}$ | $\{M, Q, K, F, I, H, A, C\}$ | 80 |
| Histone H2A and Histone H2B | $\{L, G, S, M, K, N, T, \underline{\textbf{Y}}, \underline{\textbf{F}}\}$ | $\{L, G, S, M, K, N, T\}$ | 78 |
| Interleukin-1 $\alpha$ and Interleukin-1 $\beta$ | $\{C, G, T, S, V, Q, A, \underline{\textbf{N}}, \underline{\textbf{P}}\}$ | $\{C, G, T, S, V, Q, A\}$ | 78 |
| Cytochrome P450 CYP3 and CYP51 | $\{H, F, G, K, A, P, \underline{\textbf{N}}\}$ | $\{H, F, G, K, A, P\}$ | 86 |
| **Globins** | | | |
| Myoglobin and Hemoglobin-$\alpha$ | $\{E, S, Y, V, K, \underline{\textbf{P}}, I, G, C, \underline{\textbf{W}}\}$ | $\{E, S, Y, V, K, I, G, C\}$ | 80 |
| Myoglobin and Hemoglobin-$\beta$ | $\{K, V, C, E, W, N, F, M, Y, \underline{\textbf{I}}\}$ | $\{K, V, C, E, W, N, F, M, Y\}$ | 90 |
| Hemoglobin-$\alpha$ and Hemoglobin-$\beta$ | $\{W, P, N, S, G\}$ | $\{W, P, N, S, G\}$ | 100 |
| **GPCRs** | | | |
| Rhodopsin-like and Glutamate-like | $\{D, Q, E, G, \underline{\textbf{M}}, \underline{\textbf{L}}\}$ | $\{D, Q, E, G\}$ | 67 |
| Secretin-like and Glutamate-like | $\{W, H, Y, V, D\}$ | $\{W, H, Y, V, D\}$ | 100 |
| Rhodopsin-like and Secretin-like | $\{W, E, M, S, V, H, Q, A\}$ | $\{W, E, M, S, V, H, Q, A\}$ | 100 |
| **Rhodopsin-like GPCRs** | | | |
| Aminergic receptors and Lipid receptors | $\{L, P, E, W, F, M, D\}$ | $\{L, P, E, W, F, M, D\}$ | 100 |
| Aminergic receptors and Peptide receptors | $\{L, F, E, M, K, \underline{\textbf{D}}, V, \underline{\textbf{R}}\}$ | $\{L, F, E, M, K, V\}$ | 75 |
| Lipid receptors and Peptide receptors | $\{P, R, G, I, W, \underline{\textbf{S}}, V\}$ | $\{P, R, G, I, W, V\}$ | 86 |

### E.5 $AFS$ comparison with random feature subsets

We sample 100 random feature subsets of the same size as the $AFS$ and train classifiers for each paralog pair. We compute the mean test classification scores across these subsets, see Table 8. The mean test AM score drops for these randomly selected amino acid subsets for all 15 paralog pairs. The mean test AM further drops if randomly selected amino acids contain only non-$AFS$ amino acids for 14 pairs.

*Reason for increase in score for $\alpha$-lactalbumin/lysozyme-C pair using non-AFS subset*: There is a marginal increase in test score (0.8%) with non-$AFS$ random subset for $\alpha$-lactalbumin/lysozyme-C. We suspect this to be due to the noisy nature of the dataset as the test set consists of unreviewed TrEMBL UniProt entries. For example: we find that one of the sequences in the test set (UniProt ID: T1PBP9) is identified as *C-type lysozyme/alpha-lactalbumin family* in UniProt. Another sequence (UniProt ID: Q5M8G0) is identified in UniProt as *Novel protein similar to lysozyme C*. Furthermore, the class imbalance in the test data is the reverse of that in the train data.

Table 8: The classification test AM scores using $AFS$ and random amino acid subset features for each paralog pair. The size of the random subsets here is the same size as the respective $AFS$. The '**Random Subset**' column shows the mean ($\pm$ standard deviation) of the test scores for 100 randomly sampled feature subsets. The '**Non-AFS random subset**' is similar, but the random features are sampled from the respective non-AFS amino acids.

| Paralog Pairs | AFS | Random subset | Non-AFS random subset |
|---|---|---|---|
| $\alpha$-Lactalbumin / Lysozyme C | 0.898 | 0.881 ($\pm$0.044) | 0.906 ($\pm$0.0) |
| Myoglobin / Hemoglobin-$\alpha$ | 0.970 | 0.927 ($\pm$0.032) | 0.824 ($\pm$0.0) |
| Myoglobin / Hemoglobin-$\beta$ | 0.936 | 0.918 ($\pm$0.022) | 0.823 ($\pm$0.0) |
| Hemoglobin-$\alpha$ / Hemoglobin-$\beta$ | 0.935 | 0.836 ($\pm$0.075) | 0.754 ($\pm$0.058) |
| Trypsin / Chymotrypsin | 0.835 | 0.773 ($\pm$0.075) | 0.724 ($\pm$0.086) |
| Tubulin-$\alpha$ / Tubulin-$\beta$ | 0.992 | 0.973 ($\pm$0.017) | 0.869 ($\pm$0.0) |
| Rhodopsin-like / Glutamate-like | 0.934 | 0.834 ($\pm$0.077) | 0.743 ($\pm$0.052) |
| Glutamate-like / Secretin-like | 0.845 | 0.705 ($\pm$0.083) | 0.670 ($\pm$0.065) |
| Rhodopsin-like / Secretin-like | 0.846 | 0.797 ($\pm$0.072) | 0.689 ($\pm$0.046) |
| Aminergic receptors / Lipid receptors | 0.843 | 0.782 ($\pm$0.076) | 0.637 ($\pm$0.038) |
| Aminergic receptors / Peptide receptors | 0.790 | 0.764 ($\pm$0.045) | 0.660 ($\pm$0.025) |
| Lipid receptors / Peptide receptors | 0.709 | 0.653 ($\pm$0.073) | 0.566 ($\pm$0.061) |
| Interleukin-1 $\alpha$ / Interleukin-1 $\beta$ | 0.985 | 0.954 ($\pm$0.038) | 0.818 ($\pm$0.030) |
| Histone H2A / Histone H2B | 0.934 | 0.899 ($\pm$0.032) | 0.791 ($\pm$0.028) |
| Cytochrome P450 CYP3 / CYP51 | 0.920 | 0.865 ($\pm$0.043) | 0.723 ($\pm$0.061) |

Figure 14 shows how the train and test AM scores vary for different feature subset sizes for each paralog pair. We see that feature subsets based on the top-$k$ Shapley values have in general higher train/test AM scores in comparison to random-$k$ feature subsets.

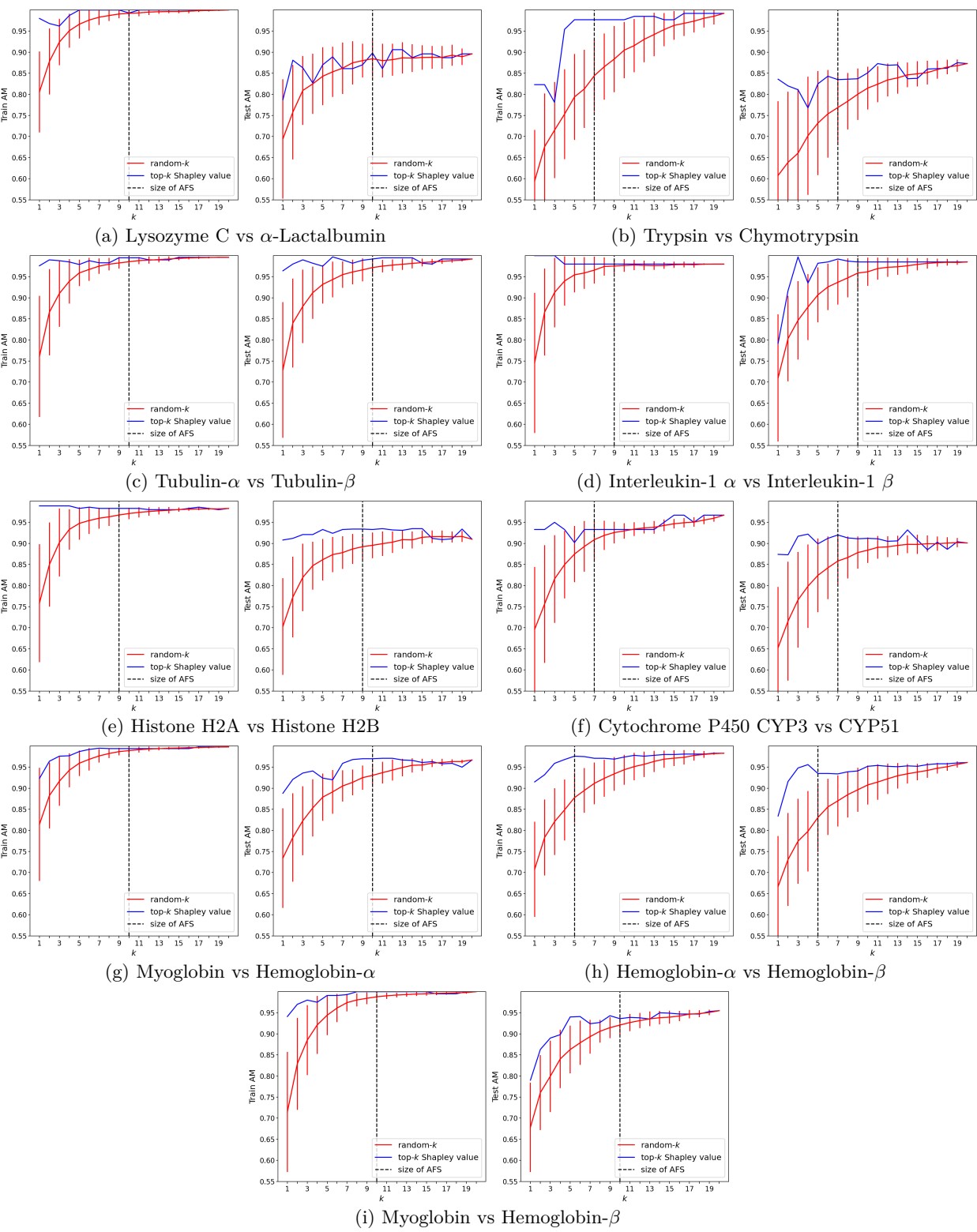

Figure 14: Classification scores (train and test AM) for feature subsets of different sizes. The blue line shows the score for top-$k$ features based on the Shapley value. The red line shows the mean ($\pm$ standard deviation) of the scores for 100 randomly selected subsets of size $k$. The train AM score here is the mean 5-fold cross-validation score on the training dataset. 

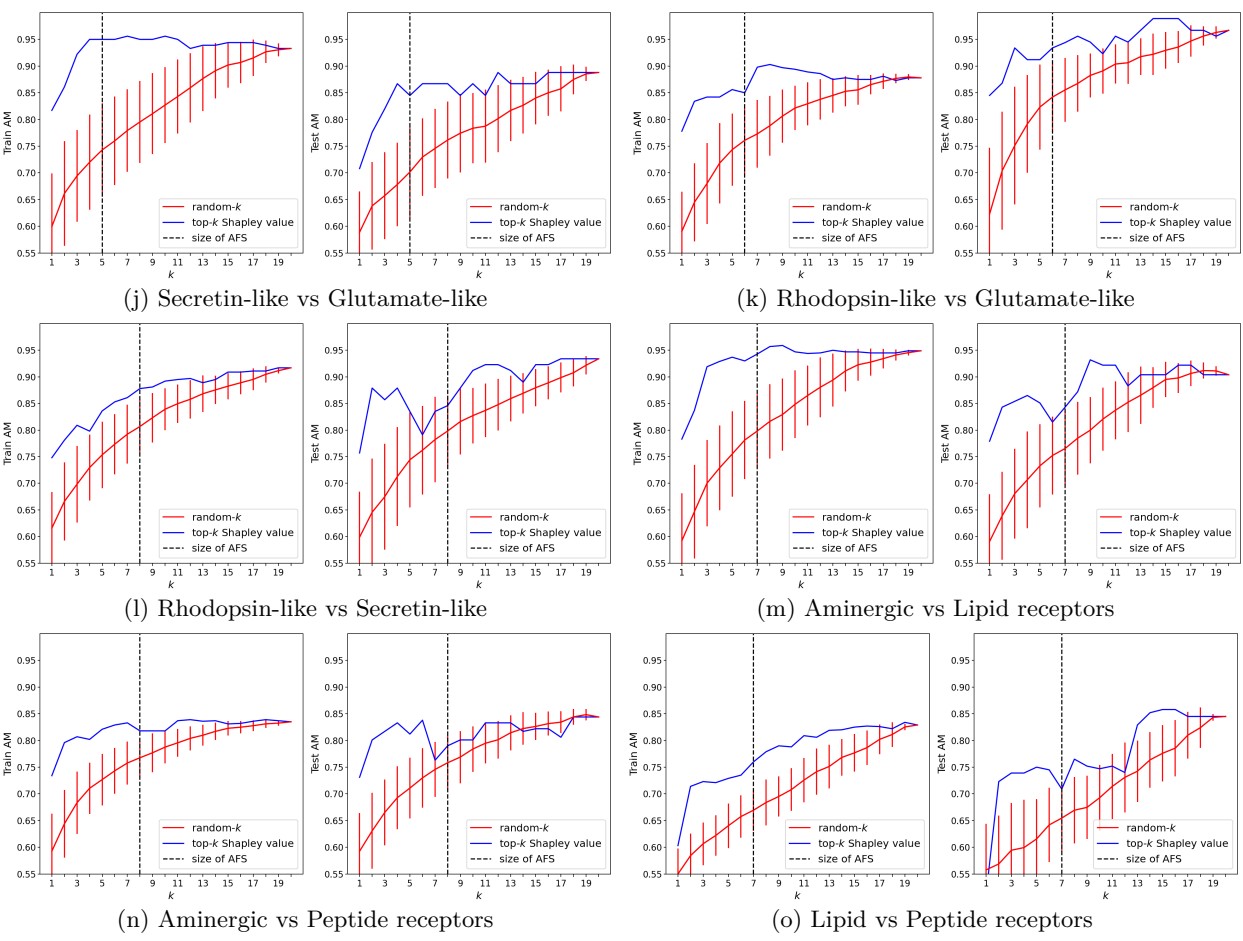

(j) Secretin-like vs Glutamate-like

(k) Rhodopsin-like vs Glutamate-like

(l) Rhodopsin-like vs Secretin-like

(m) Aminergic vs Lipid receptors

(n) Aminergic vs Peptide receptors

(o) Lipid vs Peptide receptors

Figure 14:  36.

