# OpenReview forum: "Identifying key amino acid types that distinguish paralogous proteins using Shapley value based feature subset selection"
_TMLR — Rejected by TMLR_

### Review · Reviewer_hiYz · 2025-06-16

**Summary Of Contributions:**

Parologous proteins are groups of proteins that have diverged in evolution and now perform distinction functions. Given two paralogous protein families, the paper's goal is to identify a set of key amino acids in the protein sequences that discriminate between the two families. A workflow for identifying these could be a useful exploratory tool for biologists that could help inform hypotheses.

The paper poses this as a feature selection problem, where a classifier is trained to distinguish between the two families and the selected features represent amino acids of interest. Then, the predictions are confirmed by cross referencing them with prior knowledge from the literature, from multiple sequence alignments, etc.

**Audience:**

No

**Claims And Evidence:**

No

**Requested Changes:**

Please explain why the current problem formulation is appropriate. How would such coarse-grained outputs (e.g., the set {Y, W}) actually be useful in providing insights for a biologist?

Are there ways to adapt your method (e.g., by inputting aligned sequences) that could give more precise outputs, e.g. that are localized to individual locations in a multiple sequence alignment?

**Strengths And Weaknesses:**

=Strengths=
The paper uses some real-world examples of paralogous families of interest to validate the workflow. If an exploratory tool were to demonstrate usefulness for these families, it may be useful for less-studies families in future research efforts.

=Weaknesses=
I have significant reservations about the overall problem formulation. Each protein is expressed as a length-20 vector of counts of amino acid types, completely ignoring the order of amino acids in the sequence. I understand the motivation for this: ML on variable length sequences is complex. However, the output of the workflow is so coarse that I struggle to see how it would be useful. There are 20 possible amino acids. If the workflow tells me that Y and W (2 out of 20) amino acids are important for distinguishing between the families, is this actionable? The validation of these predictions takes observations from MSAs or the literature that is more precise/localized and coarsens them to be at the resolution of amino acid types. Yes, this may be used to validate the outputs of your workflow, but using a coarse-grained comparison that is significantly less precise than the information sources being drawn from.

---

> ### Author Response · Authors · 2025-07-29
> **Responses to Reviewer hiYz**
>
> # Problem formulation
> The problem formulation focuses on the following two questions:
>  -  Does the coarse-grained information of amino acid composition carry a strong enough signal for discriminating paralogs?
>  - If so, which of the 20 amino acids have a potential role in the functional difference of the paralogs?
>
> Our results demonstrate that amino acid composition alone, without positional information, is indeed rich enough to distinguish many paralogous protein pairs. The proposed ML pipeline using the SVEA algorithm successfully identifies small, influential subsets of amino acids, and their functional relevance is consistently validated through domain-based evidence. These are also validated using classification scores on a test data and an alternate feature ranking method, MCI ([ICML, 2021](https://proceedings.mlr.press/v139/catav21a.html)).
>
> To provide clarity on the aim of the work, We have revised the manuscript to include the above research questions  (See Intro. para 2, highlighted in cyan).
>
>
> # Use in providing insights for biologists
> (As already discussed in,  Sec. 4, Conclusion) Identification of $AFS$ can be used as an initial data-driven step before doing detailed experimental investigations, like site-directed mutagenesis, to resolve sequence-function relationships. Our experiments suggest that amino acids with high Shapley values are more likely to play a role in the functional difference between the paralogs and, hence, can be targeted for site-directed mutagenesis experiments. As the size of $AFS$ is small (5-10 amino acids of 20), significantly fewer mutations can be tried.
>
> Furthermore, it is computationally cheap and requires less data. It yields quick results based on which biologists can conduct detailed experiments which are resource-intensive (time, cost, trained manpower, etc.). Its low computational requirements make it accessible to biologists with access to limited computational resources.
>
> The overall study also leads to an open question for biologists: *How is amino acid composition alone able to discriminate paralogs?*
>
>
> # Getting localised output
> Alignment-based or $k$-mer based features could be used to get more localised position-specific outputs. However, we have intentionally designed our study to test a more fundamental hypothesis: *whether the simple, coarse-grained information of amino acid composition alone carries enough signal to distinguish paralogs*.
>
> Also, position-specific features are high-dimensional, the Monte Carlo based approximation algorithm for Shapley values would require exponentially more sampling (in number of features) for good approximations. Using these features would make the computation very expensive, undermining one of our project's core goals: to provide a computationally cheap pipeline for biologists. This is already discussed in Sec. 4 Conclusion.
>
>
> # Audience
> *AI for sciences.* We believe this work demonstrates a strong use case of interpretable ML for biological sciences.

---

> > ### Comment · Reviewer_hiYz · 2025-07-31
> >
> > I continue to be skeptical of the paper's setup. In the prior case studies used for analyzing paralogs, position-wise analysis was performed. Here, the claim is that by first doing position-independent analysis a researcher could identify a set of 5-10 residues that would then need to be further explored in experiments to identify mutations driving the fundamental/mechanistic differences between paralogs. I agree that you could get a 2-4x saving from a site-directed mutagenesis scan by restricting to 5-10 residues, but this would also require exploring every possible position. An alternative approach, that identified important positions, could enable much lower-throughput experiments.
> >
> > I understand the thesis "Does the coarse-grained information of amino acid composition carry a strong enough signal for discriminating paralogs". However, if the coarse-grained information does in fact have that signal, what can you do effectively with it?

---

> > > ### Author Response · Authors · 2025-08-05
> > > **On the setup, position-based features, and the usefulness of the thesis**
> > >
> > > # The setup
> > > Our setup has the following components:
> > >  - A data-driven task that is biologically meaningful: identifying key features that distinguish pairs of paralogous proteins
> > >  - Interpretable features: amino acid composition, AAC  (20-dimensional)
> > >  - Algorithm:
> > >    - SVEA algorithm to identify the important features ($AFS$)
> > >    - Linear SVM for class-wise $AFS$ partition
> > >  - Validation:
> > >    - Primary: via domain-based evidence -  (a) multiple sequence alignment, and/or (b) 3D structure analysis, and/or (c) supporting evidence from biology literature.
> > >    - Secondary: test classification scores and alternate feature ranking method, MCI.
> > >
> > > The choice of features, AAC, above is based on two-fold reasons:
> > >  - To answer the question: Does the coarse-grained information of amino acid composition carry a strong enough signal for discriminating paralogs?
> > >  - Low-dimensional: for fast and accurate computation of Shapley values in the SVEA algorithm
> > >
> > >
> > > # Position-based features
> > > We use composition-based features; however, in principle, different choices of features can be used, including features that encode position information. Our ML pipeline with composition-based features is a fast and easy data-driven step for gathering quick information ($AFS$) before detailed analysis and experimentation.  However, as already discussed in our previous response (and Sec. 4 Conclusion) position-based features can be high-dimensional and computationally expensive for Shapley value computation / MCI. A position encoding feature engineering and a suitable algorithm for feature ranking with these high-dimensional features is worthy of a separate study in itself and directions for future work (also alluded to in Sec. 4 Conclusion), and we leave it as future work.
> > >
> > > # Use of the thesis
> > > >  ``Does the coarse-grained information of amino acid composition carry a strong enough signal for discriminating paralogs?"
> > >
> > > We believe that our thesis addresses a well-defined scientific question and thus has value in itself. Furthermore, (as discussed in Sec. 4 Conclusion, 2nd para) it posits an interesting question to biologists: *‘how amino acid composition by itself is able to distinguish paralogs, given ample evidence that 3D structure and function are conserved despite sequence divergence [(Lau et al., 2015)](https://www.sciencedirect.com/science/article/pii/S1931312814004235)’*.
> > >
> > > > ‘if the coarse-grained information does in fact have that signal, what can you do effectively with it?’
> > >
> > > An application of  $AFS$ is in hypothesis-driven investigation. For example, in globins, by analysing the $AFS$, we hypothesised that amino acids $K, E$, and $I$ are important for distinguishing the monomeric myoglobin from the tetrameric hemoglobin. This data-driven hypothesis was subsequently validated by structural analysis, which showed these specific residues are less numerous at the actual contact points of the hemoglobin tetramer and more numerous at the corresponding locations in myoglobin. See Fig. 4 and Sec. 3.1.7 Globins - Structural Analysis.
> > >
> > > This idea, that $AFS$ can identify residues critical to quaternary structure, is further corroborated by similar findings for other heteromeric proteins like Tubulin $\alpha$ / $\beta$ (Sec. 3.1.3) and Histones H2A/H2B (Sec. 3.1.4). These results demonstrate that the $AFS$ can be used prospectively to form data-driven hypotheses about the roles of specific amino acid types in lesser-studied proteins.

---

> > > > ### Comment · Reviewer_hiYz · 2025-08-05
> > > >
> > > > Thanks. It appears that the general argument is that the coarse-grained position-independent information is useful to help pose hypotheses for follow-up lines of inquiry and that AFS would not scale to a position-specific representation of amino acids. This scalability argument is not true for all feature selection methods, however. Wouldn't a simple position-specific approach be to fit a linear classification model (such as binary logistic regression) on position-specific one-hot features (feature dim = aligned_seq_len * 20) and look for features that with high absolute value in their weight? Given that AFS is not novel, I am not evaluating this paper based on details of AFS. I am just interpreting it as a black-box feature selection method.

---

> > > > > ### Author Response · Authors · 2025-08-07
> > > > > **On using position-specific features with scalable feature selection**
> > > > >
> > > > > # Feature selection quality and scalability trade-off
> > > > >
> > > > > There is a trade-off between the scalability of the feature selection method and its quality of feature importance:
> > > > >  - SVEA/MCI computes feature importance scores based on game-theoretic principles that quantify the feature importance based on the marginal contribution of a feature to multiple feature coalitions. However, it is not scalable to a large number of features.
> > > > >  - Faster ways of computing feature importance, like coefficients of logistic regression, can be of low quality, particularly for high-dimensional datasets. High-dimensional datasets are often known to have correlated features or multicollinearity, i.e. linear relation among two or more variables (Reference: Alin, 2010, https://wires.onlinelibrary.wiley.com/doi/full/10.1002/wics.84). It is a well-studied phenomenon that regression coefficients are not reliable as feature importance measures when there is multicollinearity in the features. References:
> > > > >     - (Midi et al, 2013) https://www.tandfonline.com/doi/pdf/10.1080/09720502.2010.10700699
> > > > >     - (Alin, 2010) https://wires.onlinelibrary.wiley.com/doi/full/10.1002/wics.84
> > > > >     - (Vatcheva et al, 2016) https://pmc.ncbi.nlm.nih.gov/articles/PMC4888898/
> > > > >     - (Haufe et al, 2014) https://www.sciencedirect.com/science/article/pii/S1053811913010914
> > > > >     - (Daoud, 2017) https://iopscience.iop.org/article/10.1088/1742-6596/949/1/012009
> > > > >
> > > > > For the suggested alignment-based high-dimensional position-encoding features, correlated and multicollinear features can be highly expected for paralogous protein sequences due to evolutionary and structural constraints.
> > > > >
> > > > > # Requirement of quality alignment for position-specific features
> > > > >
> > > > > The suggested position-specific one-hot features are dependent on high-quality multiple sequence alignment (MSA). MSAs are often not reliable for large and diverse protein classes like GPCRs. The protein sequences of the GPCR paralog pairs have low within-class and inter-class similarities, as illustrated in Appx. Fig. 5 (j)-(o), page 22. This results in poor quality of sequence alignment with a large number of gaps. Finding important amino acid positions in such examples can be difficult using sequence alignment. Improving the alignment quality requires tuning multiple hyperparameters like substitution matrix, gap open penalty, gap extension penalty, tree construction algorithm, etc. Thus, position-specific one-hot features for such examples can be of poor quality.
> > > > >
> > > > > # Summary
> > > > > To summarise, the efficiency of scalable feature selection methods often comes at the cost of reduced feature quality and reliability, especially for high-dimensional features.  Also, the suggested position-specific features depend on high-quality alignment, which can be expensive to obtain for protein families with low sequence similarities.
> > > > >
> > > > > The discussion on scalable feature selection methods with position-specific features digresses from the main motives of our work, i.e.,
> > > > >  - Investigating the richness of information in the amino acid composition of proteins for distinguishing paralog pairs.
> > > > >  - Identifying the key amino acid types that are important for distinguishing the paralog pairs.
> > > > >  - Providing a quick and cheap data-driven pipeline to biologists for identifying these key amino acids.
> > > > >
> > > > >  We thank the reviewer for the questions, as the discussed aspects could be useful for interested readers.

---

> > > > > > ### Comment · Reviewer_hiYz · 2025-08-07
> > > > > >
> > > > > > I agree that correlations among mutations in evolutionary data can lead to difficulties in determining the important/causal mutations that indicate the real distinction among paralogs. This seems like a fundamental challenge of this data, and it is the basis of huge amounts of attention in the statistical genetics field. Can you explain why your approach helps circumvent this? I'm assuming that even the 20-dimensional analysis may be subject to confounders from evolution.

---

> > > > > > > ### Author Response · Authors · 2025-08-12
> > > > > > >
> > > > > > > We do not suggest that the use of 20-dimensional features is to circumvent the issue of correlated features. As already answered in the previous response, the reason for the choice of composition features is:
> > > > > > >  - To answer the question: Does the coarse-grained information of amino acid composition carry a strong enough signal for discriminating paralogs?
> > > > > > >  - Low-dimensional: for fast and accurate computation of Shapley values in the SVEA algorithm
> > > > > > >
> > > > > > > The top-selected features, $AFS$, using Shapley value, are primarily validated using domain knowledge. However, to account for possible correlations in the features, we also validate the $AFS$ in our experiments using the MCI feature rankings ([Catav et al, 2021](https://proceedings.mlr.press/v139/catav21a.html)). The MCI method, by design, was aimed at addressing the issue of correlated features. [Catav et al, 2001](https://proceedings.mlr.press/v139/catav21a.html), show that Shapley value-based feature rankings are altered when correlated features are introduced. Therefore, they propose the MCI method using a different set of axioms to circumvent the issue of correlated features. Thus, we validate the Shapley value-based feature rankings with those by MCI. We find strong agreement in the two rankings (see Sec 3.3 and Appx Table 6), suggesting a lack of correlated features for the datasets considered in our experiments. We use Shapley value instead of MCI in our main pipeline for computing $AFS$, as the efficiency axiom of the Shapley value allows us to draw a fair cutoff line for selecting the important features. Since MCI doesn’t satisfy the efficiency axiom, the same cutoff line may not be applicable to it.
> > > > > > >
> > > > > > > We have now revised Sec. E.3 MCI for $AFS$ in the manuscript to include the above details.

---

### Review · Reviewer_tc7L · 2025-06-18

**Summary Of Contributions:**

This paper proposes a computationally efficient and interpretable machine learning pipeline to identify key amino acid types that distinguish pairs of paralogous proteins. The central contributions can be summarized as follows:

1. Novel Use of Shapley Value-Based Feature Subset Selection
The study applies the Shapley Value-based Error Apportioning (SVEA) algorithm to amino acid composition (AAC) features to extract a small, highly informative subset of amino acid types, termed the Amino acid Feature Subset (AFS), for each pair of paralogs.

The Shapley values are interpreted as the contribution of each amino acid type to the linear separability of the protein classes.

2. Low-Dimensional and Efficient Representation
The method identifies small subsets (5 to 10 out of 20) of amino acids that distinguish paralogous proteins effectively using only the AAC (frequency-based, order-agnostic) representation.

This results in a computationally inexpensive and interpretable alternative to deep sequence models.

3. Biological Validation Using Domain Knowledge
The biological significance of the identified AFS is validated through:

* Literature evidence citing prior wet-lab studies linking the amino acids to function.
* Multiple sequence alignment (MSA) to verify conservation within families.
* 3D structural analysis to assess positional significance in protein complexes (for example, heterodimers and tetramers).

4. Logical Consistency Across Paralog Triplets
For paralog triplets (for example, globins, GPCRs), the paper demonstrates consistency and non-overlap in AFS across all pairwise combinations, supporting biological relevance and model robustness.

5. Classification Performance
Using only the identified AFS, linear SVM classifiers achieve 70 to 99 percent accuracy on test sets across 15 protein family pairs, with most scoring above 90 percent.

6. Agreement with Independent Feature Importance Method
The identified AFS aligns closely with top-ranked features from the Marginal Contribution Feature Importance (MCI) method, validating the SVEA-derived feature ranking.

7. Disease Associations
Several identified amino acids are linked to known disease-causing mutations, suggesting potential utility in variant prioritization and experimental targeting.

**Audience:**

Yes

**Claims And Evidence:**

Yes

**Requested Changes:**

### **Recommendations for Improvement**

- Consider exploring sequence-aware features (e.g., k-mer frequencies, motif embeddings) to complement the AAC representation and potentially capture more nuanced differences.
- Extend the current binary framework to accommodate multi-class classification tasks, particularly in protein families with multiple functionally distinct subgroups.
- Quantify the variance or confidence in Shapley value estimates across Monte Carlo runs to provide a measure of stability in the AFS.
- Clarify the biological interpretation of the directionality of SVM weights associated with AFS amino acids, particularly about family-specific enrichment.
Investigate the integration of 3D visualization tools to map AFS residues onto protein structures, thereby facilitating their adoption by experimentalists.

**Strengths And Weaknesses:**

### **Strengths**

1. **Conceptual Clarity and Interpretability**
   The proposed approach is both conceptually clear and interpretable. By leveraging amino acid composition (AAC) features and a Shapley value-based feature selection algorithm (SVEA), the authors provide a framework that yields compact and biologically meaningful subsets of amino acids (AFS) capable of distinguishing paralogous protein pairs.

2. **Computational Efficiency**
   The method is computationally lightweight and avoids the use of deep learning models, making it accessible to practitioners in computational biology who lack access to high-performance computing resources. The reliance on a low-dimensional feature space (20 amino acids) permits efficient Shapley value estimation through Monte Carlo sampling.

3. **Robust and Multi-faceted Biological Validation**
   The significance of the identified amino acid subsets is substantiated through multiple complementary validation strategies, including:
   - Literature evidence supporting functional roles of the amino acids,
   - Multiple sequence alignment (MSA) to examine conservation patterns,
   - Structural analysis to assess the spatial relevance of the residues in 3D protein complexes.
   These validations enhance the biological credibility of the results.

4. **Logical Coherence Across Protein Families**
   The method exhibits consistent patterns across paralog triplets, with logical non-overlap and shared residues evident in pairwise AFS comparisons. This provides further evidence of robustness and biological plausibility.

5. **Empirical Performance**
   The classification results on test sets using only AFS features are strong, with arithmetic mean (AM) scores ranging from 70% to 99% across 15 paralog pairs. Furthermore, there is high concordance between the AFS and independently computed Marginal Contribution Feature Importance (MCI) rankings, which lends support to the reliability of the feature selection process.

6. **Practical Relevance**
   The study has high practical utility. The identified residues may serve as starting points for experimental validation via site-directed mutagenesis. In some cases, the residues are already implicated in disease phenotypes, indicating potential clinical significance.

### **Weaknesses**

1. **Lack of Positional Context**
   The methodology is based solely on amino acid composition and does not account for positional or sequence-order information. While composition is a useful global descriptor, it may obscure critical spatial or sequential motifs relevant to protein function or interaction.

2. **Restriction to Binary Classification**
   The framework is currently restricted to pairwise classification between paralogs. It remains unclear how well the approach generalizes to multi-class settings or to comparisons involving more than two related proteins simultaneously.

3. **Absence of Ground Truth for Benchmarking**
   The lack of an established ground truth for key discriminative amino acids in paralogous proteins limits the ability to perform quantitative benchmarking. While validation is extensive and convincing, it is primarily qualitative and indirect.

4. **Potential Sensitivity to Class Imbalance**
   Although the authors use class-balanced hinge loss to address imbalanced datasets, the presence of class imbalance in several protein pairs raises concerns about the reliability and generalizability of the performance metrics. Additional discussion or sensitivity analysis would be beneficial.

5. **Heuristic Nature of AFS Cutoff**
   The selection of the Shapley value cutoff for AFS inclusion is data-driven and grounded in the efficiency axiom, but its biological relevance is not guaranteed. The biological interpretability of this threshold could be strengthened by incorporating empirical or domain-specific priors.

6. **Limited Scope Beyond Paralog Discrimination**
   The study focuses exclusively on paralog pairs. It would be valuable to understand whether the methodology could be extended to other protein relationship types, such as orthologs or isoforms, or to functional prediction more broadly.

---

> ### Author Response · Authors · 2025-07-29
> **Responses to Reviewer tc7L**
>
> # Sequence-aware features / lack of positional context
> $k$-mer features can be explored to get complementary insights into sequence order information. However, we have intentionally designed our study to test a more fundamental hypothesis: *whether the simple, coarse-grained information of amino acid composition alone carries enough signal to distinguish paralogs.*
>
> Furthermore, $k$-mer based features are high-dimensional ($20^k$-dimensional). The Monte Carlo based approximation algorithm for Shapley values would require exponentially more sampling (in the number of features) for good approximations. This will make the method computationally expensive, undermining one of our project's core goals: *to provide a computationally cheap pipeline for biologists.* This is already discussed in Sec. 4, Conclusion.
>
> # Multi-class classification
> As our goal is to identify the amino acid types that play a role in the functional difference of paralogous protein pairs, and not classification per se, we use a binary framework. For protein families with multiple functionally distinct subgroups, the binary framework has been used in a one-vs-one manner, as in Sec. 3.1.7 Globins and 3.1.8 GPCRs.
>
> # Variance in Shapley value estimates
> We have now computed the Shapley value $AFS$ across 10 Monte Carlo runs. We include a section in the revised appendix with this analysis, see page 32 Sec. E.4.
>
> We do not find significant variation in these estmiates across the different runs as seen in the now included Figure 13 (page 33). (Figure can also be viewed here: [link](https://anonymous.4open.science/r/AFS_AAC_SVM-F3D9/sv_10_mc_runs.png))
>
> We do not find substantial differences in the $AFS$s across different runs. As a stronger check of consistency, for each paralog pair, we computed the intersection of the $AFS$s across the 10 runs, see the now added Table 7 (page 32). We find that for each paralog pair, this intersection of $AFS$s recovers at least 66.6\% (two-thirds) of the amino acids in the (single-run) $AFS$ reported in the paper. For 14 pairs, greater than 75\% is recovered. For 4 pairs, 100\% is recovered. Furthermore, the amino acids with the top-3 Shapley values reported in the paper are present in the intersection of $AFS$s for all paralog pairs. For 13 pairs, the top-5 amino acids are in the intersection.
>
>
> # Interpretation of the directionality of SVM weights
> The linear SVM has a weight $w$ corresponding to each amino acid type. For a given amino acid, the sign (+ve/-ve) indicates which class in general has a greater composition of that amino acid. This is discussed in page 4, Sec. 2.3 Protein family-wise partition of $AFS$ using SVM.
>
> # 3D visualisation
> We can include a link to a Google Colab notebook where users can provide a PDB ID (unique identifier for each structure in the [Protein Data Bank](https://www.rcsb.org/)), along with the identified $AFS$, to get an output as shown below. To maintain anonymity only a screenshot of the Colab notebook is provided here: [link](https://anonymous.4open.science/r/AFS_AAC_SVM-F3D9/colab_structure_visualisation.png).
>
> # On sensitivity to class imbalance
> As already discussed in Sec. 2.4 - *Validation of $AFS$* (*Using test data*, page 5), we use the arithmetic mean of sensitivity and specificity (AM) to measure the performance of the classifier. This measure is not biased towards the classifier’s performance on the majority class.
>
> Furthermore, we do not find significant sensitivity to class imbalance in our computational experiments. This is illustrated by the classification scores on the test sets for paralog pairs with a difference in train and test class-imbalance ratios. For example,
>  - lysozyme C/$\alpha$-lactalbumin has class ratios 1:3.4 in the train set and 3.8:1 in the test set, i.e., the class-imblance is reversed in the train and test sets.
>  - Myoglobin/Hemoglobin-$\beta$ has class ratios 1:2.8 in the train set and 1:1.1 in the test set, i.e., class-imbalance is decreased in the test set.
>
> Such differences in train/test class imbalances are present for many other paralog pairs, see Appx. Table 2 (page 17).

---

### Review · Reviewer_LEg4 · 2025-07-16

**Summary Of Contributions:**

The submission presents a machine learning pipeline that uses Shapley value-based feature selection (SVEA) on amino acid composition (AAC) features to identify a small subset of amino acid types (termed AFS) that differentiate pairs of paralogous proteins. The key contributions are: (i) applying SVEA to protein AAC data to extract interpretable features, (ii) partitioning the selected amino acids by class using linear SVM weights, and (iii) validating the biological relevance of these amino acids via literature, multiple sequence alignment, and structural data. The work demonstrates that a small number of amino acid types (5–10) can achieve classification performance comparable to the full AAC representation and that these amino acids often correspond to known functionally important residues. The proposed approach is computationally inexpensive and potentially useful as a pre-screening step for guiding wet-lab experiments.

The claim that the proposed method is computationally inexpensive holds primarily because it operates on a very low-dimensional feature space—just the 20 amino acid types—without considering **positional information**. This simplicity enables efficient Shapley value computation. However, this advantage disappears if more expressive features are used. For example, incorporating k-mer frequencies or position-specific features (e.g., from alignment or structural data) would drastically increase the dimensionality, making exact or even approximate Shapley value computation **computationally impractical**. Thus, the tractability of the approach hinges entirely on the coarse granularity of the AAC representation.

**Audience:**

Yes

**Claims And Evidence:**

No

**Requested Changes:**

# Comments

**1 — Implicit aim**
Your narrative implicitly positions SVEA + AAC as a *biologist-friendly screening tool*: the goal is to pick a handful (≈ 5-10) of amino-acid types that both (i) separate two paralogous families almost as well as the full 20-dimensional AAC representation and (ii) line up with existing structural, mutational and disease evidence. In other words, you claim that “Shapley-ranked AAC features are a reliable proxy for functional specificity”.

**2 — Missing random-subset baseline**
Because the search space is tiny ( $\binom{20}{5\text{–}10}$ ) and every AAC feature is dense, many random 5- or 10-amino-acid subsets will inevitably carry signal. A simple experiment that draws N random subsets of the same size and reports the distribution of test AM would show how much “lift” SVEA actually provides over chance; at the moment readers cannot tell whether your gains are statistically meaningful.

**3 — Missing comparison to standard ML feature selectors**
No results are reported for off-the-shelf selectors such as Random-Forest Gini importance, mutual-information ranking, univariate $t$-tests, L1-SVM/LASSO, or SHAP on an RF or XGB model. Without that, it is unclear whether SVEA is better, comparable, or worse than widely-used, computationally cheaper alternatives.

**4 — Missing comparison to biologically motivated amino acid selection**
A straightforward, biologically grounded baseline would be to rank amino acid types by their **average conservation score across MSA columns** (e.g. using column entropy or any conservation metric), and select the top 5–10 most conserved amino acids. This approach is simple, intuitive, and widely accepted among biologists as a proxy for functional relevance. Comparing your Shapley-based AFS selections to this conservation-based ranking would offer a meaningful benchmark: is SVEA doing better than what a domain expert might reasonably infer from standard alignment analysis alone?


**5 — AAC vs AFS not convincingly analysed**
Appendix Table 5 contains both AAC and AFS test AM scores, but the paper never discusses them. A quick scan shows AAC beating AFS in **9 of 15 datasets (\~60 %)**, with the remainder a near-draw. That weakens any claim of superior predictive power for AFS and instead suggests that its benefit lies chiefly in interpretability or dimensionality reduction. An explicit analysis (e.g. a critical-difference diagram) is needed.



###  **Critical Adjustments**

1. **Add a random subset baseline comparison (Comment #2)**
   Include an experiment that compares the performance of SVEA-selected AFS against randomly selected amino acid subsets of the same size (e.g. 5–10). Report the distribution of test AM scores across random trials.
   **→ Critical**, as this directly tests whether SVEA provides meaningful lift over chance in such a small feature space.

2. **Compare to standard ML feature selection methods (Comment #3)**
   Benchmark SVEA against established selectors like:

   * Random Forest feature importance
   * Mutual information
   * Univariate $t$-test
   * LASSO or L1-SVM
   * SHAP values from tree-based models
     **→ Critical**, as without this, the added value of SVEA over standard alternatives is unproven.

3. **Explicitly analyze AAC vs AFS performance (Comment #5)**
   Include a detailed comparison of AAC vs AFS classification results from Appendix Table 5 in the main text. Consider using a critical-difference diagram or a summary table.
   **→ Critical**, since this addresses whether feature selection provides any benefit in predictive performance.


4. **Quantify and visualize the tradeoff between interpretability and predictive performance**
   The current manuscript assumes a sweet spot around 5–10 selected amino acids, but provides no systematic analysis of how predictive performance varies with the number of features selected. To clarify this tradeoff, please plot **test AM as a function of the number of top-ranked amino acids** (according to your SVEA ranking), sweeping from 1 up to all 20. This curve would reveal:

   * Whether there is a sharp drop-off below a certain feature count,
   * Whether the full AAC feature set significantly improves performance,
   * And how many features are needed before performance plateaus.

   This analysis would make explicit the balance between model simplicity (interpretability) and classification accuracy, and help justify your default choice of 5–10 amino acids per AFS.
   **→ Critical**, as it would enhance transparency and strengthen the case for AFS as a practical tool.




###  **Recommended Adjustments**

5. **Add a conservation-based feature selection baseline (Comment #4)**
   Use MSA-derived conservation scores (e.g. column entropy averaged per amino acid type) to rank amino acids and select the top-k most conserved. Compare test AM performance with SVEA-selected AFS.
   **→ Recommended**, as this would contextualize SVEA against a simple, biologically intuitive baseline widely used in practice.

6. **Clarify the scope and limits of the method's efficiency (Related to Comment #1)**
   Acknowledge that the low computational cost of Shapley value computation is a direct result of the fixed 20-dimensional AAC space. Highlight that richer representations (e.g. k-mers, position-specific features) would make the approach infeasible.
   **→ Recommended**, for intellectual honesty and to better position the method's use case.

**Strengths And Weaknesses:**

###  **Strong Aspects**

1. **Clear biological motivation and relevance**:
   The paper targets an important and well-motivated problem—identifying distinguishing amino acid features between paralogous proteins—in a way that is interpretable and potentially useful to experimental biologists.

2. **Computational simplicity**:
   By using amino acid composition (AAC) and a small, fixed feature space (20 dimensions), the approach remains computationally efficient and easy to implement, which makes it accessible to non-specialist users.

3. **Interpretability of results**:
   The use of Shapley values allows for transparent feature selection, and the downstream partitioning via SVM weights adds class-specific interpretability.

4. **Comprehensive biological validation**:
   The authors support their findings with literature, sequence conservation (MSA), and structural analysis across 15 diverse paralog pairs, which strengthens the biological credibility of their selected features.

5. **Potential as a pre-screening tool**:
   The proposed pipeline could serve as a low-cost filter for prioritizing residues in site-directed mutagenesis or functional annotation studies.

---

###  **Weaker Aspects (Requiring Attention)**

1. **Lack of comparison to a random subset baseline**:
   Given the small search space and dense features, random subsets of amino acids could also yield decent performance. Without a random baseline, the added value of Shapley-based selection remains unclear.

2. **No comparison to standard ML feature selection methods**:
   The paper omits comparisons with well-established alternatives like Random Forest importance, mutual information, univariate tests, LASSO, or SHAP. These could serve as important sanity checks or stronger baselines.

3. **No biologically motivated baseline (e.g., conservation)**:
   A simple yet powerful alternative—ranking amino acid types by their average conservation score across MSA columns—is not tested. This would be a natural benchmark for biologists and an important point of comparison.

4. **AAC vs AFS comparison not highlighted or analyzed**:
   While Appendix Table 5 contains results for both AAC and AFS, the paper does not analyze them. AAC outperforms AFS in 9 of 15 cases (\~60%), suggesting AFS does not offer superior predictive power. The authors should explicitly analyze this and clarify the tradeoff.

5. **Scalability depends entirely on feature simplicity**:
   The efficiency of the method hinges on using 20-dimensional AAC features. The approach would become computationally infeasible if position-specific features (e.g., k-mers or sequence motifs) were used—this limitation should be clearly acknowledged.

---

> ### Author Response · Authors · 2025-07-29
> **Responses to Reviewer LEg4**
>
> # Comment 1:  Implicit aim
> This work investigates the following two questions:
>  - Does the coarse-grained information of amino acid composition carry a strong enough signal for discriminating paralogs?
>  - If so, which of the 20 amino acids have a potential role in the functional difference of the paralogs?
>
> Our results demonstrate that amino acid composition alone, without positional information, is indeed rich enough to distinguish many paralogous protein pairs. The proposed ML pipeline using the SVEA algorithm successfully identifies small, influential subsets of amino acids, and their functional relevance is consistently validated through domain-based evidence. The computational efficiency of the ML pipeline makes it a biologist-friendly screening tool.
>
> The revised manuscript now includes these research questions to clarify our aim. See Intro para 2 in cyan.
> # CA (critical adjustment) 1: random subset baseline comparison
> The manuscript is revised with a new subsection in the appendix (Sec. E.5 *$AFS$ comparison with random feature subsets*, page 34). The new Table 8 shows that, for all 15 paralog pairs, the test AM score of the SVEA-selected $AFS$ is significantly higher than the mean score of 100 randomly selected subsets of the same size. The performance gap is even larger when random subsets are drawn only from non-AFS amino acids (for 14 of 15 pairs).
> # CA 2: comparison to standard ML feature selectors
> We chose SVEA because it is an axiomatic method based on cooperative game theory and considers interaction between features for quantifying their importance. The cooperative effect of different amino acids can be important for a protein’s function and hence useful in distinguishing paralogs.
> ## Off-the-shelf feature selectors
>  - Mutual information and univariate t-test ignore feature interactions.
>  - Lasso/L1-SVM finds a sparse classifier but doesn't rank all features by importance.
>  - Random Forest Gini importance doesn't explicitly or fairly attribute interaction effects.
>  - (As discussed in page 2, penultimate para) SHAP explains a trained model's prediction of a *test* data point. It is not a method for computing feature importance at a dataset level. While SVEA looks at the inter-class linear separation at the dataset level for different feature subsets for computing feature importance.
> ## Comparison with MCI
> We compare the SVEA with MCI ([ICML, 2021](https://proceedings.mlr.press/v139/catav21a.html)), which is a relevant method. This is another axiomatic method that considers feature interactions with similar computational complexity. As discussed in Sec. 3.3 and Appx. E.3, the results show strong agreement, validating our findings.
> # CA 3: On predictive power of $AFS$
> ## Comment 5
> > ‘weakens any claim of superior predictive power (compared to AAC)  for AFS ... suggests that its benefit lies chiefly in interpretability or dimensionality reduction’
>
> We do not claim superior predictive power of $AFS$ in comparison to AAC. However, as there is no significant drop in the test scores with $AFS$, it validates the importance of these amino acids. We consider classification scores on the test data as secondary validation in comparison to our primary validation via domain-based ground truth evidence; therefore, the classification scores are pushed to the appendix.
> # CA 4-5
> ## ‘assumes a sweet spot around 5–10 selected amino acids’
> The $AFS$ size is determined by a data-driven cutoff based on the Shapley value efficiency axiom. The cutoff, in principle, can be user-defined as it is only used to select the top-ranked features. The cutoff selects the top-ranked features that have above-average Shapley values. See Sec. 2.2 last para.
> ## ‘no .. analysis of how predictive performance varies with the number of features’
> The focus of the paper is not driven towards achieving higher predictive performance, but to suggest the best set of amino acids for wet-lab experimentalists to focus on. Thus, we did not discuss classification performance using different sizes of top-ranked amino acids. However, for interested readers, we include plots for the classification score trend with the number of features in the revised appendix. See Appx. E.5 Fig. 14 page 35.
> # RA 5: MSA conservation based feature selection
> Our goal is to evaluate the potential of using only the composition feature without position information. Features based on MSA conservation that use the position information of the sequence deviate from this goal. Hence, we do not discuss such features. However, we discuss the conservation of $AFS$ amino acids for 7 paralog pairs in their respective MSAs (See Fig. 2 and Appx. Fig. 9-12).
> # RA 6: Acknowledging scope and limits of the method's efficiency
> ## low computational cost due to 20-dimensions and infeasibility with higher dimensions
> This is already discussed and acknowledged in Sec. 4 Conclusion para 2.

---

### Decision · Action_Editor_sv7b · 2025-08-26

**Recommendation:** Reject

**Additional Comments:**

If you plan a major revision, please address the following to meet TMLR’s acceptance criteria:

- Empirical baselines and controls:

1. Add comparisons to standard feature selection methods (e.g., LASSO with stability selection, univariate tests, Random Forest importance, aggregated SHAP to obtain global importances). Report both selection overlap and downstream classification with the selected subsets.
2. Promote the random-subset experiment into the main text: summarize distributions over many random draws with mean±std, confidence intervals, and significance testing (e.g., permutation test or bootstrap), plus visualizations (box/violin plots).
3. Include a conservation-based biological baseline: compute per–amino-acid conservation statistics from the same MSAs used for validation, rank amino acids, and compare top‑k sets to SVEA (overlap metrics, enrichment analyses).

- Trade-offs and performance:

1. Move the AAC vs AFS comparison into the main paper. Provide a clear analysis of performance deltas and discuss when AFS is preferable (e.g., interpretability, data scarcity, downstream wet‑lab feasibility), even if accuracy is slightly lower.
2. Include accuracy vs feature-count curves with interpretation, linking empirical trends to the theoretically motivated cutoff (efficiency axiom) and justifying the 5–10 feature range.

- Robustness and statistics:

1. Add uncertainty quantification (multiple runs, different random seeds, or cross-validation folds), with confidence intervals for key metrics. 2. Report variance in Shapley estimates and resulting subset stability across runs (e.g., Jaccard overlap).

- Scope and scientific utility:

1. Clarify how location-agnostic amino-acid selection can concretely aid hypothesis generation. If possible, provide case studies where selected amino acids connect to known functional mechanisms, or demonstrate a follow-up positional analysis seeded by the AFS.
2. Discuss computational scalability and, if feasible, include a variant with higher-dimensional features (e.g., k‑mer composition, physicochemical grouped features) using simpler selectors, to show the method’s relevance beyond the 20‑feature space.

- Presentation:

1. Surface key experimental summaries from the appendix into the main text with clear figures/tables and concise takeaways.
2. Ensure that code and data pipelines fully reproduce all new analyses.

**Audience:**

Yes

**Audience Explanation:**

TMLR’s audience includes researchers interested in applying ML to scientific discovery and interpretability in biology. The idea of using Shapley-based subset selection to identify amino acid types that discriminate paralog pairs, with cross-checks via MSA/3D structure/literature, will likely interest a subset of readers focused on interpretable, hypothesis-generating models in computational biology. The application domain is timely, and the pipeline is reproducible and potentially useful as a preliminary screening tool; provided stronger empirical positioning and benchmarking.

**Claims And Evidence:**

No

**Claims Explanation:**

Per TMLR’s evaluation criteria, acceptance hinges on whether claims are supported by accurate, convincing, and clear evidence. While the manuscript is methodologically sound in applying an existing Shapley value–based subset selection (SVEA) to amino acid composition (AAC), several core empirical supports remain insufficient:

- **Missing or underdeveloped baselines**: There is no empirical comparison against standard feature selection alternatives (e.g., LASSO, univariate tests, Random Forest importance, aggregated SHAP), despite the low-dimensional setting making such comparisons straightforward. The authors’ theoretical distinctions do not preclude benchmarking; without it, it is unclear whether SVEA offers practical advantages.

- **Random subset control**: The newly added random-subset comparison is relegated to the appendix, without distributional summaries, variance, or significance testing. This weakens the claim that SVEA reliably outperforms chance in this 20-feature space.

- **AAC vs AFS trade-off**: The paper acknowledges that the full AAC representation outperforms the selected AFS in most pairs (9/15), but this central trade-off (interpretability vs performance) is not analyzed in the main text. Readers lack guidance on when AFS is preferred and why.

- **Biological plausibility baseline**: There is no comparison to a simple, biologically meaningful conservation-based ranking (e.g., per–amino-acid conservation derived from MSAs already used for validation). Such a baseline would help contextualize whether selected features align with domain expectations.

- **Granularity and utility**: The problem granularity (selecting 5–10 amino acid types (location-agnostic)) may be too coarse to support strong biological hypothesis generation. Reviewers’ concerns about scientific usefulness at this level are not fully addressed by the current evidence.

- **Statistical reporting**: Most results are presented without uncertainty quantification, making it difficult to judge robustness across random seeds or resampling.

Given these gaps, the current evidence does not yet convincingly establish that SVEA-derived amino-acid subsets provide added value beyond simpler baselines or chance, nor that conclusions are robust across runs.

**Resubmission Of Major Revision:**

The authors may consider submitting a major revision at a later time.